# Adversarial Attacks on Black Box Video Classifiers: Leveraging the Power of Geometric Transformations

**Shasha Li**,* **Abhishek Aich**\*, **Shitong Zhu, M. Salman Asif, Chengyu Song,**
**Srikanth V. Krishnamurthy, Amit K. Roy-Chowdhury**
University of California, Riverside, CA, USA

## Abstract

When compared to the image classification models, black-box adversarial attacks against video classification models have been largely understudied. This could be possible because, with video, the temporal dimension poses significant additional challenges in gradient estimation. Query-efficient black-box attacks rely on effectively estimated gradients towards maximizing the probability of misclassifying the target video. In this work, we demonstrate that such effective gradients can be searched for by parameterizing the temporal structure of the search space with geometric transformations. Specifically, we design a novel iterative algorithm Geometric TRAnsformed Perturbations (GEO-TRAP), for attacking video classification models. GEO-TRAP employs standard geometric transformation operations to reduce the search space for effective gradients by searching for a small group of parameters that define these operations. This group of parameters describes the geometric progression of gradients, resulting in a reduced and structured search space. Our algorithm inherently leads to successful perturbations with surprisingly few queries. For example, adversarial examples generated from GEO-TRAP have better attack success rates with $\sim 73.55\%$ fewer queries compared to the state-of-the-art method for video adversarial attacks on the widely used Jester dataset. Overall, our algorithm exposes vulnerabilities of diverse video classification models and achieves new state-of-the-art results under black-box settings on two large datasets. Code is available here: https://github.com/sli057/Geo-TRAP

## 1 Introduction

Adversarial attacks are designed to expose vulnerabilities of Deep Neural Networks (DNNs). With real-world applications of video classification based on DNNs emerging [1–3], a key question that arises is "*what type of adversarial inputs can mislead, and thus render video classification networks vulnerable*?" Designing such adversarial attacks not only helps expose security flaws of DNNs, but can also potentially stimulate the design of more robust video classification models.

Adversarial attacks against image classification models have been studied in both *white-box* [4–8] and *black-box* [9–13] settings. In the white-box setting, an adversary has full access to the model under attack, including its parameters and training settings (hyper-parameters, training data, etc.) In the black-box setting, an adversary only has partial information about the victim model, such as the predicted labels of the model. In the case of video classification models, adversarial attacks in both white-box and black-box settings have garnered some interest [14–22], although the body of work here is more limited than the case of image classification models.

A common black-box attack paradigm is query-based, wherein the attacker can send queries to the victim model to collect the corresponding predicted labels, and thereby estimate the gradients needed

---

*Equal contribution. Corresponding author: Shasha Li (sli057@ucr.edu)

35th Conference on Neural Information Processing Systems (NeurIPS 2021).

Table 1: **Comparison with state-of-the-art.** GEO-TRAP, compared to current black-box attack methods for videos, doesn't train a different network to craft perturbations, and parameterizes the temporal dimension of videos in searching for effective perturbation directions.

| Methods | WITHOUT training a "perturbation" network | CONSIDER temporal dimension? | PARAMETERIZE temporal dimension? |
|---|---|---|---|
| PATCHATTACK[19] | ✗ | ✗ | ✗ |
| HEURISTICATTACK [20] | ✓ | ✗ | ✗ |
| SPARSEATTACK [21] | ✗ | ✗ | ✗ |
| MOTION-SAMPLER ATTACK [22] | ✓ | ✓ | ✗ |
| GEO-TRAP (**Ours**) | ✓ | ✓ | ✓ |

for curating the adversarial examples. Unlike static images, videos naturally include additional information from the temporal dimension. This high dimensionality (i.e., sequence of frames instead of one image) poses challenges to black-box adversarial attacks against video classification models; in particular, significantly more queries are typically needed for estimating the gradients for crafting adversarial samples [19–22]. [19] reduces the number of queries by adding perturbations on the patch level instead of at the pixel level; [20, 21] propose to add perturbations only on key pixels. [22] considers the intrinsic differences between images and videos (i.e., the temporal dimension), and proposes to use the optical-flow of clean videos as the motion prior for adversarial video generation. Similar to [22], we also explicitly consider the temporal dimension of video. However, rather than fixing the temporal search space using the motion prior of clean videos, we propose to parameterize the temporal structure of the space with geometric transformations. This results in a better structured and reduced search space, which allows us to generate successful attacks with much fewer queries in black-box settings than the state-of-the-art methods, including [22].

**Contributions.** In this paper, we propose a novel query-efficient black-box attack algorithm against video classification models. Due to the extra temporal dimension, generating video perturbations by searching for effective gradients remains a challenging task given the exceedingly large search space. These gradients are estimated by searching for 'directions' that maximize the probability of the victim model mis-classifying the crafted inputs. Our approach drastically reduces this large search space by defining this space with a small set of parameters that describe the geometric progression of gradients in the temporal dimension, resulting in a reduced and temporally structured search space. Conceptually, this parameterization of the temporal structure of the search space is performed using geometric transformations (e.g. affine transformations). We refer to our algorithm as Geometrically TRAnsformed Perturbations, or GEO-TRAP. Despite this surprisingly simple strategy, GEO-TRAP outperforms existing black-box video adversarial attack methods by significant margins ($\sim 1.8\%$ improvement in attack success rate with $\sim 73.55\%$ fewer queries for targeted attacks in comparison to the state-of-the-art [22] on the Jester dataset [23]).

## 2 Related Works

In this section, we review different black-box adversarial attacks strategies, and categorize our proposed method with respect to state-of-the-art black-box attacks designed for video classifiers.

In most real-world attacks, the adversary only has partial information about the victim models, such as the predicted labels. In such black-box settings, the adversary can first attack a local surrogate model and then transfer these attacks to the target victim model [24, 25], formally called as *transferability-based* black-box attack. Alternatively, they may estimate the adversarial gradient with zero-order optimization methods such as Finite Differences (FD) or Natural Evolution Strategies (NES) by querying the victim model [12, 13, 26], which is called *query-based* black-box attack. GEO-TRAP falls under the category of query-based black-box attacks (designed for videos).

Whilst several white-box attacks have been proposed for video classification models [14–18], black-box video attacks are relatively under explored. PATCHATTACK (V-BAD)[19] is the first to propose a black-box video attack framework which uses a hybrid attack strategy of first generating initial perturbations for each video frame by attacking a local image classifier, and then updating the perturbations by querying the victim model. Compared to PATCHATTACK[19] , GEO-TRAP does not require training a local classifier. PATCHATTACK[19] crafts video perturbations by treating each frame as a separate image, but reduces the search space of the gradient estimation by morphing the

perturbations in patches/partitions. However, its attack performance has been shown to be inferior to that of a more recent approach [22] (discussed below). HEURISTICATTACK [20] uses a query-based attack strategy, and reduces the search space by generating adversarial perturbations only on heuristically selected key frames and salient regions. SPARSEATTACK [21] reduces the search space by adding perturbations only on key frames using a reinforcement learning based framework. MOTION-SAMPLER ATTACK [22] proposed a query-based attack strategy that utilizes a motion excited sampler to obtain *motion-aware* perturbation prior by using the optical-flow of the clean video. This motion-aware prior reduces the search space for gradients resulting in fewer queries. Similar to [22] but different from [19–21], GEO-TRAP explicitly considers the temporal dimension of video in order to search for effective gradients. However unlike [22], GEO-TRAP does not fix the temporal structure of the search space using a *pre-computed fixed* motion prior, but parameterize it with simple geometric transformations. These black-box video attack methods are summarized in Table 1.

## 3    Attacking via Geometrically TRAnsformed Perturbations (GEO-TRAP)

**Notation.** We denote the tuple of a video clip and its corresponding label as $(\boldsymbol{x}, y)$, which represents a data-point in the distribution $\mathcal{X}$. Each video sample $\boldsymbol{x} \in \mathbb{R}^{T \times H \times W \times C}$ has $T$ frames of $H$ height, $W$ width, and $C$ channels. We denote the victim video classification model as $\boldsymbol{f_\theta} : \mathcal{X} \to \mathcal{Y}$, where $\boldsymbol{\theta}$ represents the model's parameters learned from the training subset of $\mathcal{X}$, via a mapping to the label space $\mathcal{Y}$. We further assume $\mathcal{X}$ consists of videos from $|\mathcal{Y}| = K$ categories. To make the perturbations imperceptible to humans, we impose the perturbation budget $\rho_{\max}$ with the $\|\cdot\|_p$ norm. Throughout this paper, we consider $\|\cdot\|_\infty$ norm following [15, 19, 22] (the method can be extended to $p = 1, 2$ norms). To constrain $\|\cdot\|_\infty$ of perturbation below a budget $\rho_{\max}$, we use the clip($\cdot$) function to keep the perturbation pixel value in $[-\rho_{\max}, \rho_{\max}]$. The function sign($\cdot$) extracts the sign of given input variable. The superscript $i$, throughout the paper, denotes the iteration $i$. The subscript $t$ denotes the frame index. For clarity, we represent vectors/tensors with the bold font and scalars with the regular font.

**Problem Statement.** We consider the scenario of attacking a standard video classification model using a query-based paradigm under **black-box settings** (assuming no access to $\boldsymbol{\theta}$ nor the training subset of $\mathcal{X}$). Specifically, we aim to craft perturbed videos $\boldsymbol{x}_{\text{adv}}$ with imperceptible differences from $\boldsymbol{x}$, in order to alter the decision of the target model $\boldsymbol{f_\theta}$ via multiple queries to guide the gradient estimation. This problem can be mathematically formulated as follows.

$$\underset{\boldsymbol{x}_{\text{adv}}}{\arg\min} \quad \mathcal{L}\big(\boldsymbol{f_\theta}(\boldsymbol{x}_{\text{adv}}), y\big) \quad \text{s.t.} \quad \|\boldsymbol{x}_{\text{adv}} - \boldsymbol{x}\|_\infty \le \rho_{\max} \tag{1}$$

$\mathcal{L}\big(\boldsymbol{f_\theta}(\boldsymbol{x}_{\text{adv}}), y\big)$ is the objective function, capturing the similarity between the classifier's output and the ground truth label $y$, and varies with different attack goals (targeted or untargeted). The challenge is to obtain $\boldsymbol{x}_{\text{adv}}$ with as few queries as possible by estimating gradient $\boldsymbol{g}^\star = \nabla_{\boldsymbol{x}_{\text{adv}}} \mathcal{L}\big(\boldsymbol{f_\theta}(\boldsymbol{x}_{\text{adv}}), y\big)$, which is unknown in the considered black-box setting.

**Overview of GEO-TRAP.** We propose a novel iterative video perturbation framework that follows the principle of the Basic Iterative Method [27] in order to fool $\boldsymbol{f_\theta}$ under $\|\cdot\|_\infty$ norm as follows.

$$\boldsymbol{x}_{\text{adv}}^{(0)} = \boldsymbol{x}, \quad \boldsymbol{x}_{\text{adv}}^{(i)} = \texttt{clip}\big(\boldsymbol{x}_{\text{adv}}^{(i-1)} - h\,\texttt{sign}(\boldsymbol{g}^{(i)})\big) \tag{2}$$

where $h$ is a hyperparameter and $\boldsymbol{g}^{(i)}$ is the gradient estimated by querying the black-box victim model at the $i^{th}$ iteration using our proposed GEO-TRAP algorithm. As shown in (2), effective perturbations rely on the guidance of the gradient $\boldsymbol{g}^{(i)}$. Therefore, efficiently estimating $\boldsymbol{g}^{(i)}$ is at the core of GEO-TRAP for successfully subverting video classifiers. We execute the following two steps in each iteration to estimate $\boldsymbol{g}^{(i)}$.

1. For any input video $\boldsymbol{x}^{(i)}$, a random noise tensor $\boldsymbol{r}_{\text{frame}} \in \mathbb{R}^{H \times W \times C}$ and a set of geometric transformation parameters $\boldsymbol{\Phi}_{\text{warp}} \in \mathbb{R}^{T \times D}$ are chosen with each element sampled from a standard normal distribution. $D$ represents the number of parameters needed for the geometric transformation of a single frame (details are provided in Section 3.2). In this setup, our search space for estimating $\boldsymbol{g}^{(i)}$ consists of $\boldsymbol{r}_{\text{frame}}$ and $\boldsymbol{\Phi}_{\text{warp}}$.

2. We then warp $\boldsymbol{r}_{\text{frame}}$ with $\boldsymbol{\Phi}_{\text{warp}}$ to get the candidate direction $\boldsymbol{\pi} = [\boldsymbol{r}_1, \boldsymbol{r}_2, \cdots, \boldsymbol{r}_T] \in \mathbb{R}^{T \times H \times W \times C}$ (see Algorithm 2.1 TRANS-WARP). $\boldsymbol{\pi}$ is then employed to compute a gradient estimator $\boldsymbol{\Delta}$ by querying the black-box victim model with a standard gradient estimation algorithm (see Algorithm 2 GRAD-EST). The gradient estimator $\boldsymbol{\Delta}$ is then used to update $\boldsymbol{g}^{(i)}$.

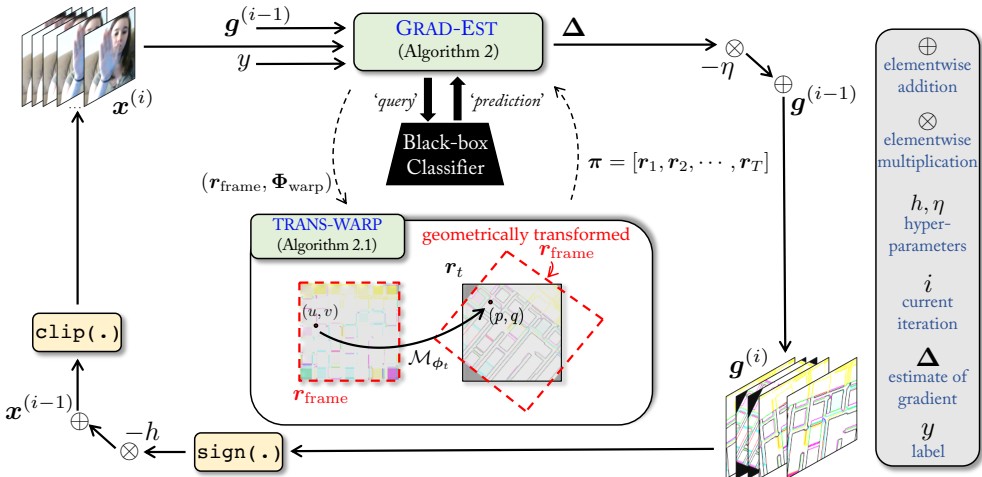

Figure 1: **Overview of GEO-TRAP.** GEO-TRAP is a black-box attack algorithm guided by the key observation that strong gradients $g^{(i)}$ can be computed by finding better gradient search direction candidates $\pi$. We propose to search each frame of the directions $r_t$ by warping a randomly sampled $r_{\text{frame}}$ using a geometric transformation $\mathcal{M}_{\phi_t}$; different $r_t$ in $\pi$ are warped by the same $r_{\text{frame}}$, thus have geometric progression among frames.

The overall attack strategy is summarized in Algorithm 1 and pictorially illustrated in Figure 1. Since in Step 2 above, the gradient estimation (i.e., GRAD-EST) procedure includes the geometric transformation strategy (i.e., TRANS-WARP), we will next describe GRAD-EST and then move on to TRANS-WARP. For the simplicity of exposition, we drop the superscript $i$ and shorten the loss function to $\mathcal{L}(f_\theta(x_{\text{adv}}), y)$ to $\mathcal{L}$ (as the model parameters $\theta$ remain unchanged) in rest of this section.

### 3.1  GEO-TRAP Gradient Estimation (GRAD-EST)

Let $g^\star = \nabla_x \mathcal{L}$ be the ideal value of the gradient of $\mathcal{L}$ at $x$, required to create $x_{\text{adv}}$ in (2). To find an efficient estimator $g$ for $g^\star$, a (new) surrogate loss $\ell(g) = -\langle g^\star, g \rangle$ is defined such that the estimator $g$ has a sufficiently large inner product with the actual gradient $g^\star$ ($g$ is normalized to a unit vector; we ignore the normalization operation for ease of explanation). The loss function definition and the algorithm to estimate $g$ follow [13].

As $g^\star$ is unknown in the black-box setting, this surrogate loss function can be estimated as

$$\ell(g) = -\langle g^\star, g \rangle = -\langle \nabla_x \mathcal{L}, g \rangle \approx -\frac{\mathcal{L}(x + \epsilon g, y) - \mathcal{L}(x, y)}{\epsilon}. \tag{3}$$

To iteratively estimate $g$, we need to, in turn, estimate the gradient of $\ell(g)$, i.e., $\Delta = \nabla_g \ell(g)$. With antithetic sampling [28], $\Delta$ can be estimated as

$$\Delta = \frac{\ell(g + \delta\pi) - \ell(g - \delta\pi)}{\delta}\pi, \tag{4}$$

where $\delta$ is a small number adjusting the magnitude of the loss variation and $\pi \in \mathbb{R}^{T \times H \times W \times C}$ is a random candidate direction. Our core contribution lies in the fact that instead of randomly sampling $\pi$ in the search space [13], we reduce the search dimensionality by warping a randomly sampled tensor $r_{\text{frame}} \in \mathbb{R}^{H \times W \times C}$ with another randomly sampled geometric (e.g., affine) transformation parameter tensor $\Phi_{\text{warp}} \in \mathbb{R}^{T \times D}$ to get $\pi$. The search space is then reduced from $T \times H \times W \times C$ to $(H \times W \times C) + (T \times D)$ and $D$ is a relatively small number, $D \ll H \times W \times C$. With $w_1 = g + \delta\pi$ and $w_2 = g - \delta\pi$ and combining (3) with (4), we get

$$\Delta = \frac{\mathcal{L}(x + \epsilon w_2, y) - \mathcal{L}(x + \epsilon w_1, y)}{\epsilon\delta}\pi. \tag{5}$$

Note that by querying the victim model $f_\theta$ with $x + \epsilon w_1$, we are able to retrieve the value of $\mathcal{L}(x + \epsilon w_1, y)$; similarly we can obtain the value of $\mathcal{L}(x + \epsilon w_2, y)$ ($\mathcal{L}(\cdot)$ is defined following [17].). In summary, we estimate $\Delta$ with these two queries to the victim model. The resulting algorithm for estimating gradient of $\nabla_g \ell$ or $\Delta$ for consequently estimating $g$ is shown in Algorithm 2. Eventually at every iteration, we use $\Delta$ to update $g$ by applying a one-step gradient descent as $g \leftarrow g - \eta\Delta$, where $\eta$ is a hyperparameter to update $g$. This updated $g$ is later used to obtain $x_{\text{adv}}$ using (2).

**Algorithm 1:** GEO-TRAP: Query-based Iterative attack for Video Classifiers

**Input** : video $\boldsymbol{x}$, corresponding label $y$, step-size $\eta$ for updating the gradient, step-size $h$ for updating adversarial video.

**Output :** adversarial video $\boldsymbol{x}_{\text{adv}}$

1 **Initialize**: $\boldsymbol{x}^{(0)} = \boldsymbol{x}, \boldsymbol{g}^{(0)} = \boldsymbol{0}, i = 1$

2 **while** $\text{argmax}\left(\boldsymbol{f}_{\boldsymbol{\theta}}(\boldsymbol{x}^{(i)})\right) = y$ **do**

3     $\boldsymbol{\Delta} = \text{GRAD-EST}(\boldsymbol{x}^{(i-1)}, \boldsymbol{g}^{(i-1)}, y)$   /\* Gradient Estimation \*/

4     $\boldsymbol{g}^{(i)} \leftarrow \boldsymbol{g}^{(i-1)} - \eta\boldsymbol{\Delta}$

5     $\boldsymbol{x}^{(i)} \leftarrow \text{clip}(\boldsymbol{x}^{(i-1)} - h\text{sign}(\boldsymbol{g}^{(i)}))$

6     $i \leftarrow i + 1$

7 **end**

8 **return** $\boldsymbol{x}_{\text{adv}} = \boldsymbol{x}^{(i)}$

### 3.2 Noise Warping using Geometric Transformation (TRANS-WARP)

To tackle the challenge of the high-dimensionality of the search space, we propose to parameterize the search space with a single random noise tensor $\boldsymbol{r}_{\text{frame}} \in \mathbb{R}^{H \times W \times C}$ and a sequence of geometric transformations $\boldsymbol{\Phi}_{\text{warp}} \in \mathbb{R}^{T \times D}$. Apart from the reduction of the search space of gradient estimation, our geometric transformation provides a temporal structure to $\boldsymbol{\pi}$, which we discuss next.

At every iteration, $\boldsymbol{\pi} = [\boldsymbol{r}_1, \boldsymbol{r}_2, \dots, \boldsymbol{r}_T]$ represents the candidate direction for $\boldsymbol{\Delta}$. These directions $\boldsymbol{r}_t \in \mathbb{R}^{H \times W \times C}$ are used to compute $\boldsymbol{\Delta}$ in order to update gradient $\boldsymbol{g}$. To obtain $\boldsymbol{\pi}$, we use a sequence of transformation vectors $\boldsymbol{\Phi}_{\text{warp}} = [\boldsymbol{\phi}_1, \boldsymbol{\phi}_2, \dots, \boldsymbol{\phi}_T]$ where $\boldsymbol{\phi}_t \in \mathbb{R}^D$. The dimensionality $D$, chosen by the attacker, can vary depending on the transformation type that is populated from $\boldsymbol{\phi}_t$, e.g., $D = 6$ for affine transformation. We take affine transformation as an example to describe the warping process. We start by randomly sampling $\boldsymbol{r}_{\text{frame}}$ and the sequence of $\boldsymbol{\phi}_t$ along with initializing each element in the sequence of $\boldsymbol{r}_t$ with zero in **every** iteration. TRANS-WARP then computes $\boldsymbol{r}_t$ by warping $\boldsymbol{r}_{\text{frame}}$ using the parameters in $\boldsymbol{\phi}_t = [\phi_{11}^t, \phi_{12}^t, \phi_{13}^t, \phi_{21}^t, \phi_{22}^t, \phi_{23}^t] \in \mathbb{R}^6$ of $\boldsymbol{\Phi}_{\text{warp}} \in \mathbb{R}^{T \times 6}$ as follows. For all $C$ channels, let $(p, q)$ and $(u, v)$ be the target and source coordinates in $\boldsymbol{r}_t$ and $\boldsymbol{r}_{\text{frame}}$, respectively. $\boldsymbol{r}_t$ (for all channels) is computed as

$$\boldsymbol{r}_t(p, q) \leftarrow \boldsymbol{r}_{\text{frame}}(u, v), \quad 1 \le p, u \le H, \ 1 \le q, v \le W. \tag{6}$$

Location $(p, q)$ is computed using the affine transform matrix $\mathcal{M}_{\boldsymbol{\phi}_t}$ created with $\boldsymbol{\phi}_t$ in homogeneous coordinates [29] as shown below. $t$ is dropped for simplicity.

$$\begin{pmatrix} p \\ q \\ 1 \end{pmatrix} = \mathcal{M}_{\boldsymbol{\phi}} \begin{pmatrix} u \\ v \\ 1 \end{pmatrix} = \begin{bmatrix} \phi_{11} & \phi_{12} & \phi_{13} \\ \phi_{21} & \phi_{22} & \phi_{23} \\ 0 & 0 & 1 \end{bmatrix} \begin{pmatrix} u \\ v \\ 1 \end{pmatrix} \tag{7}$$

We compactly denote this warping operation in (6) and (7) with $\boldsymbol{r}_t = \mathcal{T}(\boldsymbol{r}_{\text{frame}}, \boldsymbol{\phi}_t)$. Affine transformation allows translation, rotation, scaling, and skew to be applied to $\boldsymbol{r}_{\text{frame}}$ to get each $\boldsymbol{r}_t$. Therefore, the sequence of $\boldsymbol{r}_t$ have affine geometric progression among its temporal dimension. Other examples of geometric transformations may be more constrained, such as the similarity transformation $\mathcal{M}_{\boldsymbol{\phi}}^{\text{S}}$ (that allows translation, dilation (uniform scale) and rotation with $D = 4$) and translation-dilation $\mathcal{M}_{\boldsymbol{\phi}}^{\text{TD}}$ (that allows translation and uniform dilation with $D = 3$) as shown below.

$$[\phi_{11}, \phi_{12}, \phi_{13}, \phi_{23}] \rightarrow \mathcal{M}_{\boldsymbol{\phi}}^{\text{S}} = \begin{bmatrix} \phi_{11} & \phi_{12} & \phi_{13} \\ -\phi_{12} & \phi_{11} & \phi_{23} \\ 0 & 0 & 1 \end{bmatrix}, [\phi_{11}, \phi_{13}, \phi_{23}] \rightarrow \mathcal{M}_{\boldsymbol{\phi}}^{\text{TD}} = \begin{bmatrix} \phi_{11} & 0 & \phi_{13} \\ 0 & \phi_{11} & \phi_{23} \\ 0 & 0 & 1 \end{bmatrix} \tag{8}$$

## 4 What Makes GEO-TRAP Effective?

Potent iterative algorithms should rely on few queries for crafting successful perturbations for time efficiency. To minimize the number of queries, iterative algorithms need to find strong gradients in their early iterations. As discussed earlier, videos inherently incur a larger search space due to the temporal dimension and thus, pose challenges in searching for effective gradients. In this section, we provide empirical evidence to show that by parameterizing the temporal dimension, GEO-TRAP finds better gradients, in general, than previous works. We use three baselines in this analysis.

**Algorithm 2:** GRAD-EST($\boldsymbol{x}^{(i-1)}, \boldsymbol{g}^{(i-1)} \in \mathbb{R}^{T \times H \times W \times C}, y) \to$ Estimate $\boldsymbol{\Delta} = \nabla_{\boldsymbol{g}}\ell(\boldsymbol{g}) \in \mathbb{R}^{T \times H \times W \times C}$

**Input**   :video $\boldsymbol{x}^{(i)}$, label $y$, gradient estimator $\boldsymbol{g}^{(i-1)}$, $\delta$ for loss variation, $\epsilon$ for approximation.
**Output:** estimation of $\boldsymbol{\Delta} = \nabla_{\boldsymbol{g}}\ell(\boldsymbol{g})$

1 **Sample** $\boldsymbol{r}_{\text{frame}} \in \mathbb{R}^{H \times W \times C}$, $\boldsymbol{\Phi}_{\text{warp}} \in \mathbb{R}^{T \times D}$ (each element from a normal distribution $\mathcal{N}(0, 1)$)
2 $\boldsymbol{\pi} = $ TRANS-WARP($\boldsymbol{r}_{frame}, \boldsymbol{\Phi}_{warp}$)   /* Use Geometric Transformations */
3 $\boldsymbol{w}_1 = \boldsymbol{g}^{(i-1)} + \delta\boldsymbol{\pi}, \boldsymbol{w}_2 = \boldsymbol{g}^{(i-1)} - \delta\boldsymbol{\pi}$
4 $L_1 = \mathcal{L}(\boldsymbol{x}^{(i-1)} + \epsilon\boldsymbol{w_2}, y), L_2 = \mathcal{L}(\boldsymbol{x}^{(i-1)} + \epsilon\boldsymbol{w_1}, y)$   /* Query victim model twice */
5 $\boldsymbol{\Delta} = (L_2 - L_1)^{\boldsymbol{\pi}}/_{\epsilon\delta}$
6 **return** $\boldsymbol{\Delta}$

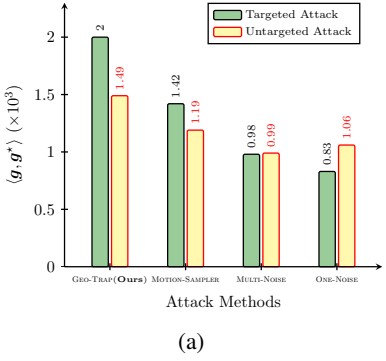
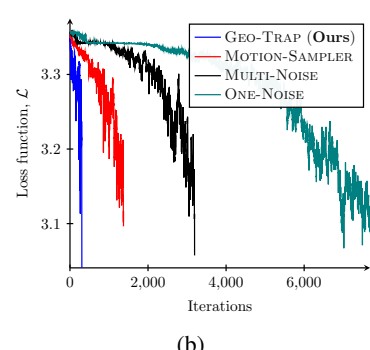

(a)                (b)

Figure 2: **Gradient Analysis of GEO-TRAP. (a)** GEO-TRAP's high query-efficiency is a direct implication of good quality gradient estimation (for both targeted and untargeted attack), shown here with higher cosine similarity with $\boldsymbol{g}^\star$ compared to other methods. **(b)** Better quality of estimated gradients by GEO-TRAP results in a successful attack with fewer queries compared to other attacks.

- MULTI-NOISE ATTACK [13] which computes search directions $\boldsymbol{r}_t$ separately for each frame by sampling each element of $\boldsymbol{r}_t$ from a standard normal distribution, resulting in a search space dimension of $T \times H \times W \times C$. It does not explicitly consider the temporal dimension; temporal progression in any arbitrary direction is possible between a sequence of perturbation frames.

- ONE-NOISE ATTACK which computes $\boldsymbol{r}_1$ by sampling each element from a standard normal distribution and applies the same $\boldsymbol{r}_1$ across all $\boldsymbol{r}_t(t = 1, 2, \cdots, T)$. ONE-NOISE ATTACK reduces the search space but completely ignores the temporal dimension when generating the perturbation.

- MOTION-SAMPLER ATTACK [22] which uses the optical flow of the original video $\boldsymbol{x}$ to warp $\boldsymbol{r}_{\text{frame}}$ to get each $\boldsymbol{r}_t$. It reduces the search space by using the motion prior of $\boldsymbol{x}$ as the temporal progression between perturbation frames. In contrast, rather than fixing the temporal search space using a motion prior, GEO-TRAP parameterizes the temporal structure of the space with $\boldsymbol{\Phi}_{\text{warp}}$.

We measure the gradient estimation quality by calculating the cosine similarity between the ground truth $\boldsymbol{g}^\star$ and the estimated gradient $\boldsymbol{g}$ following [19] for the aforementioned baselines. For each attack, we average over 1000 randomly selected videos with their cosine similarity values in the first attack iteration. We choose the first iteration because the initial $\boldsymbol{g}^\star$ is the same for the different attack methods, ensuring a fair comparison. As shown in Figure 2(a), our proposed method for estimating the gradients, yields $\boldsymbol{g}$ of the best quality for both untargeted and targeted attacks among all evaluated approaches. This leads to faster loss convergence / few queries as shown in Figure 2(b). We validate such trends with different loss functions and more datasets in the Supplementary Material.

The empirical results validate that by carefully considering the temporal dimension and parameterizing the temporal structure of the gradient search space with geometric transformations, GEO-TRAP finds better gradients. GEO-TRAP and MOTION-SAMPLER ATTACK [22] are better than the other two; the reason could be that temporally structured perturbations are more likely to disrupt the motion context of videos. However, the gradients estimated by MOTION-SAMPLER ATTACK [22] are not as effective as our proposed approach; the reason could be that the motion-prior of the clean video does not necessarily represent the temporal behavior of effective video perturbations. By allowing flexibility of the temporal progression while maintaining only a minimally sufficient space through its geometric parameterization, GEO-TRAP generates effective temporally structured perturbations. Note that one could use other, potentially better ways to parameterize the temporal progression of the video perturbation; this is regarded as future works.

**Algorithm 2.1:** TRANS-WARP($r_{\text{frame}} \in \mathbb{R}^{H \times W \times C}$, $\Phi_{\text{warp}} \in \mathbb{R}^{T \times D}$)$\rightarrow$ Estimate $\pi \in \mathbb{R}^{T \times H \times W \times C}$

---

**Input** :noise tensor $r_{\text{frame}}$, warp tensors $\Phi_{\text{warp}}$, transformation operation $\mathcal{T}_\phi(\cdot)$.
**Output**:candidate directions $\pi = [r_1, r_2, \cdots, r_T]$.
1 **Initialize** $\pi = \emptyset$
2 **for** $t = [1, 2, \cdots, T]$ **do**
3 $\quad$ $\phi_t = \Phi_{\text{warp}}[t]$
4 $\quad$ $r_t = \mathcal{T}(r_{\text{frame}}, \phi_t)$ $\quad$ /* Warping Operation */
5 $\quad$ $\pi \leftarrow$ append $r_t$
6 **end**
7 **return** $\pi$

---

## 5  Experiments

**Datasets.** Following previous work like [15], we use the human action recognition dataset UCF-101 [30] and the hand gesture recognition dataset 20BN-JESTER (Jester) [23] to validate our attacks. *UCF-101* includes 13320 videos from 101 human action categories (e.g., applying lipstick, biking, blow drying hair, cutting in the kitchen). Given the diversity it provides, we consider the dataset to validate the feasibility of our attacks on *coarse-grained* actions. *Jester*, on the other hand, includes hand gesture videos that are recorded by crowd-sourced workers performing 27 kinds of gestures (e.g., sliding hand left, sliding two fingers left, zooming in with full hand, zooming out with full hand). The appearance of different hand gestures is similar; it is the motion information that matters in the video classification. We use this dataset to validate our attack with regard to *fine-grained* actions.

**Baselines.** Among the four state-of-the-art black-box video attack methods [19–22] described in Section 2, we use [21, 22] as baselines for following reasons. Our first baseline is MOTION-SAMPLER ATTACK [22], which has been shown to outperform PATCHATTACK[19] , ONE-NOISE and MULTI-NOISE attacks (introduced in Section. 4). Our second baseline is HEURISTICATTACK [20]. We note that SPARSEATTACK [21] is not included in our analysis as we couldn't replicate their results.

**Attack Settings.** We consider four state-of-the-art video classification models representing diverse methodologies of learning from videos, i.e., C3D [31], SlowFast [32], TPN [33] and I3D [34], as our black-box victim models to attack. More details about the four video models are provided in the Supplementary Materials. For UCF-101, we randomly select one video from each category following the setting in [19, 22]. For Jester, since the number of categories is small, we randomly select four videos from each category. All attacked videos are correctly classified by the black-box model. For targeted attack, a random target class is chosen for each video. The maximum noise value $\rho_{\max}$ is 10 pixel values (out of 255) following [15, 35, 36]. We provide more results for different $\rho_{\max}$ in Supplementary Material. Note that since the perturbation generated by HEURISTICATTACK [20] is sparse and thus more imperceptible, we do not impose a perturbation budget on it. We set the maximum query limit to $Q = 60,000$ for untargeted attack and $Q = 200,000$ for targeted attack. The other hyper-parameters, i.e., $\epsilon$, $\delta$, $\eta$, and $h$ take the same values as mentioned in [22]. Unless otherwise specified, a translation-dilation transformation (with $D = 3$) is used for our attack method.

**Metrics.** Following [19, 22], we evaluate GEO-TRAP, in terms of *(a)* Success Rate (SR), i.e., the total success rate in attacking within query and perturbation budgets; and *(b)* Average Number of Queries (ANQ) i.e., the average total queries from attacks for all videos (including failed ones).

### 5.1  Comparison to State-of-the-Art

**Untargeted Attack.** We report the untargeted attack performance of our attack method and the baseline methods in Table 2. We observe that, in general, GEO-TRAP requires fewer average number of queries when attacking different black-box victim models: on average over 45 % fewer queries than MOTION-SAMPLER ATTACK [22]. At the same time, GEO-TRAP yields higher attack success rates: on average about 6% higher than HEURISTICATTACK [20]. When attacking SlowFast model on the Jester dataset, GEO-TRAP achieves 100% successful rate with only 521 queries while the baseline methods need at least 1906 queries. We also observe that the TPN model is more robust towards black-box attacks compared with the other three video recognition models.

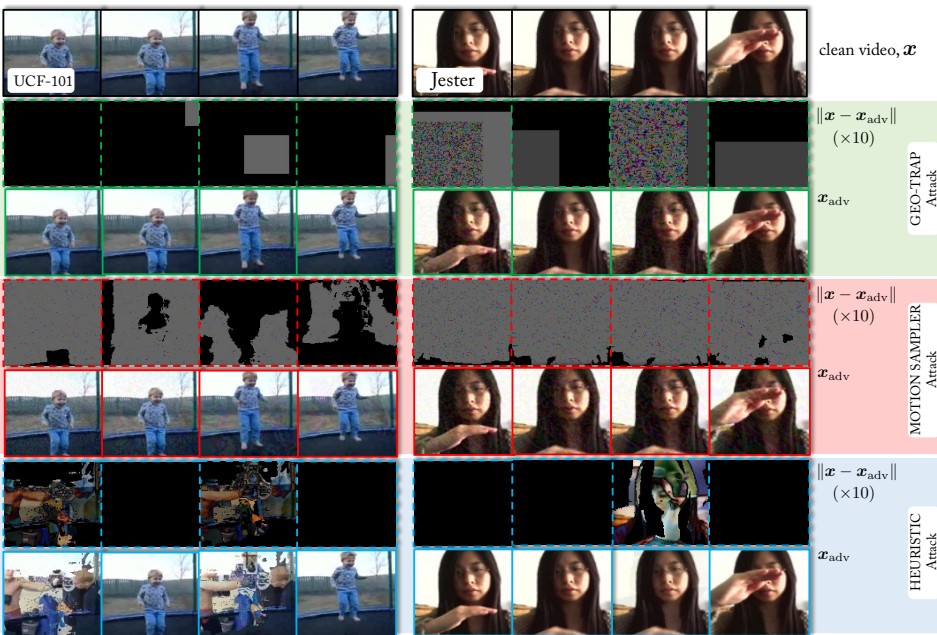

Figure 3: **Visualization of Perturbations and Perturbed Video.** We visualize the generated perturbations and perturbed video for GEO-TRAP and other baselines for UCF-101 (*left*) and Jester (*right*) datasets for untargeted attack against SlowFast classifier with $\rho_{max} = {}^{10}/_{255}$.

**Visualization.** We show two visualizations of adversarial frames on Jester and UCF-101 in Fig. 3. We observe that the generated adversarial frames have little difference from the clean ones but can lead to a failed classification. Also, our attack method could lead to *sparse* perturbations in the spatial and temporal dimension as the perturbations are sometimes zoomed out (thus get very small), and sometimes are translated out of the sight with choice of geometric transformation. More examples are in the Supplementary Material.

**Targeted Attack.** We report the targeted attack performance of our method and the baseline methods in Table. 3. We observe that in some cases, HEURISTICATTACK [20] requires fewer number of queries than GEO-TRAP, but its attack success rates are pretty low in those cases. For example, when attacking the TPN model on the Jester dataset, although HEURISTICATTACK [20] requires only 12k average number of queries, its attack success rate is less than half of ours, 44.4% v.s. 92.6%. GEO-TRAP consistently yields higher attack success rates, on average over 30% higher than HEURISTICATTACK [20] and over 8% higher than MOTION-SAMPLER ATTACK [22]. In addition, in most cases, GEO-TRAP requires fewer average number of queries than the two baseline attacks, on average over 45 % fewer queries than MOTION-SAMPLER ATTACK [22]. The targeted attack performance further validates the effectiveness of our method.

## 5.2 Different Geometric Transformations in TRANS-WARP

As discussed in Section 3.2, different kinds of geometric transformations could be used in the TRANS-WARP function. In addition to the translation-dilation transformation ($\mathcal{M}_\phi^{TD}$ in (8), $D = 3$) employed throughout the paper, we report the performance of GEO-TRAP with two other different geometric

Table 2: **Untargeted Attacks**. GEO-TRAP demonstrates highly successful untargeted attacks (high Success Rate (SR)) with fewer queries (low Average Number of Queries (ANQ))

| Datasets | Methods | Black-box Video Classifiers | | | | | | | |
| | | C3D | | SlowFast | | TPN | | I3D | |
| | | ANQ (↓) | SR (↑) | ANQ (↓) | SR (↑) | ANQ (↓) | SR (↑) | ANQ (↓) | SR (↑) |
|---|---|---|---|---|---|---|---|---|---|
| Jester | HEURISTICATTACK [20] | 4699 | 99.0% | 3572 | 98.1% | 4679 | 82.0% | 4248 | 98.1% |
| | MOTION-SAMPLER ATTACK [22] | 4549 | 99.0% | 1906 | 100% | 6269 | 91.3% | 3029 | 99.4% |
| | GEO-TRAP (**Ours**) | 1602 | 100% | 521 | 100% | 3315 | 92.4% | 1599 | 100% |
| UCF-101 | HEURISTICATTACK [20] | 5206 | 70.2% | 3507 | 87.2% | 6539 | 71.8% | 6949 | 84.7% |
| | MOTION-SAMPLER ATTACK [22] | 14336 | 81.6% | 4673 | 97.2% | 20369 | 75.8% | 7400 | 94.4% |
| | GEO-TRAP (**Ours**) | 11490 | **86.2%** | 1547 | **98.8%** | 17716 | **76.1%** | 4887 | **97.4%** |

Table 3: **Targeted Attacks**. GEO-TRAP demonstrates highly successful targeted attacks (high Success Rate (SR)) with fewer queries (low Average Number of Queries (ANQ))

| Datasets | Methods | Black-box Video Classifiers | | | | | | | |
|---|---|---|---|---|---|---|---|---|---|
| | | C3D | | SlowFast | | TPN | | I3D | |
| | | ANQ ($\downarrow$) | SR ($\uparrow$) | ANQ ($\downarrow$) | SR ($\uparrow$) | ANQ ($\downarrow$) | SR ($\uparrow$) | ANQ ($\downarrow$) | SR ($\uparrow$) |
| Jester | HEURISTICATTACK [20] | 15595 | 46.3% | 30768 | 98.1% | **12006** | 44.4% | 31088 | 77.8% |
| | MOTION-SAMPLER ATTACK [22] | 26704 | 98.2% | 33087 | 100% | 63721 | 80.9% | 39037 | 90.7% |
| | GEO-TRAP (**Ours**) | **6198** | **100%** | **7788** | **100%** | 41294 | **92.6%** | **19542** | **98.2%** |
| UCF-101 | HEURISTICATTACK [20] | **26741** | 29.0% | 22152 | 61.4% | **71828** | 36.4% | 92244 | 43.7% |
| | MOTION-SAMPLER ATTACK [22] | 100467 | 71.1% | 57126 | 86.0% | 151409 | 31.6% | 96498 | 59.6% |
| | GEO-TRAP (**Ours**) | 71820 | **85.8%** | **21878** | **95.0%** | 141629 | **40.0%** | **76708** | **74.6%** |

transformations, i.e., similarity transformation ($\mathcal{M}_\phi^S$ in (8), $D = 4$) and affine transformation ($\mathcal{M}_\phi$ in (7), $D = 6$). Figure 4 shows the untargeted attack performance on Jester with these different geometric transformations (more results are available in the Supplementary Material). We observe that the transformation with fewer degrees of freedom (DOF) ( translation-dilation transformation) tends to require fewer queries while having the same or higher attack success rates (the attack success rates are available in the Supplementary Material). We believe that $D = 3$ provides enough temporal flexibility to disrupt the motion context of the videos; additional degrees of freedom seemingly increase the search space unnecessarily, resulting in more queries.

## 6    Conclusion

Black-box adversarial attacks on video classifiers is a challenging problem that has been largely understudied. In this work, we demonstrate that searching for effectual gradients in a reduced but structured search space for crafting perturbations leads to highly successful attacks with fewer queries compared to state-of-the-art attack strategies. In particular, we propose a novel iterative algorithm that employs Geometric transformations to parameterize and reduce the search space, for estimating gradients that maximize the probability of mis-classification of the perturbed video. This simple and novel strategy exposes the vulnerability of widely used video classifiers. For instance, GEO-TRAP decreases average query numbers by 64.78%, 72.66% and 47.21% to attack C3D, SlowFast, and I3D, respectively, for almost 100% success rate in untargeted attacks.

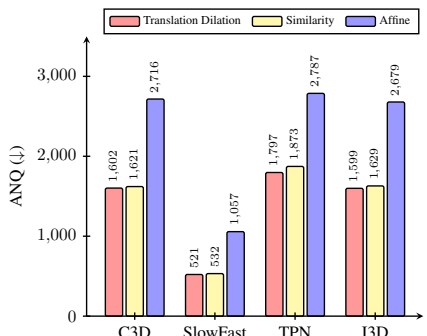

Figure 4: **Performance with different $\mathcal{M}_\phi$**. GEO-TRAP results in best performance when $\mathcal{M}_\phi$ is set as translation-dilation operation.

## 7    Broader Impact

In this work, by leveraging geometric transformations for effective gradient estimations, we propose a highly query-efficient adversarial attack on video classification models which demonstrates state-of-the-art results. As more and more safety-critical systems (e.g., perceptual modules in autonomous vehicles) nowadays rely on video models, we are hopeful that our work, in addition to future research (including designing sophisticated video generative models as in [37–39] for distribution-driven attacks [40, 41]), can eventually help build sufficiently robust video models to best avoid malicious sub-versions. On one hand, we believe our algorithm could allow further research in adversarial robustness and data augmentation strategies of deep vision models. It should also give a direction to researchers to design counter defense methodologies [42–46]. On the other hand, it highlights a key drawback of different video classifiers which will allow adversaries to design more sophisticated attacks, both in white-box and black-box settings. Addressing such fallacies in designing deep neural networks is of utmost importance before introducing them in real-world scenarios.

**Acknowledgement.** The authors would like to thank Dr. Cliff Wang of US Army Research Office for his extensive comments and input on this work. This material is based upon work supported by the Defense Advanced Research Projects Agency (DARPA) under Agreement No. HR00112090096. Approved for public release; distribution is unlimited.

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
