# Adversarial Attacks on Black Box Video Classifiers: Leveraging the Power of Geometric Transformations (Supplementary Material)

Shasha Li[1], Abhishek Aich[1], Shitong Zhu, M. Salman Asif, Chengyu Song,
Srikanth V. Krishnamurthy, Amit K. Roy-Chowdhury
University of California, Riverside, CA, USA

## CONTENTS

## List of Tables

## List of Figures

---

[1]Equal contribution. Corresponding author: Shasha Li (`sli057@ucr.edu`)

## A    Victim Video Classifiers: Clean Test Accuracy

We consider four state-of-the-art video classification models, representing diverse methodologies of learning from videos, i.e., C3D [1], SlowFast [2], TPN [3] and I3D [4], as our black-box victim models to perform adversarial attack. The *C3D* model applies 3D convolution to learn spatio-temporal features from videos. *SlowFast* uses a two-pathway architecture where the slow pathway operates at a low frame rate to capture spatial semantics and the fast pathway operates at a high frame rate to capture motion at fine temporal resolution. *TPN* captures actions at various tempos by using a feature-level temporal pyramid network. *I3D* proposes the Inflated 3D ConvNet(I3D) with Inflated 2D filters and pooling kernels of traditional 2D CNNs. All the models are trained using open-source toolbox MMAction2 [5] with their default setups. The test accuracy of the victim models with clean 16-frame videos on both UCF-101 and Jester datasets are shown in Table 1. Note that both datasets do not contain personally identifiable information and offensive contents.

Table 1: Clean test Accuracy of the victim classifiers

| Datasets | Black-box Video Classifiers | | | |
|---|---|---|---|---|
| | C3D | SlowFast | TPN | I3D |
| UCF-101 | 78.8% | 85.4% | 74.3% | 71.7% |
| Jester | 90.1% | 89.5% | 90.5% | 91.2% |

## B    Additional Experiments with Different Perturbation Budgets $\rho_{\max}$

We present additional analysis of the attack performance of GEO-TRAP and our two baseline methods, i.e., HEURISTICATTACK [6] and MOTION-SAMPLER ATTACK [7] for $\rho_{\max} = 8, 16$ in Table 2. Note that for comprehensibility, we also provide the results for $\rho_{\max} = 10$ from the main manuscript in Table 2. We observe that GEO-TRAP consistently outperforms MOTION-SAMPLER ATTACK [7]; GEO-TRAP requires less number of queries while achieves same or higher attack success rates.

## C    Statistical Comparison of Different Attack Methods

We have three sources of randomness in our experiments: *a*) the sampling of $r_{\text{frame}}$ in both GEO-TRAP and MOTION-SAMPLER ATTACK [7] and the sampling of $\Phi_{\text{warp}}$ in GEO-TRAP; *b*) direction initialization sampling in HEURISTICATTACK [6]; *c*) target label sampling in targeted adversarial attacks for all three methods. To account for all these three randomness, we run the targeted attack against I3D model on Jester dataset under perturbation budget $\rho_{\max} = 16$ for the three methods for five times. Using targeted attack strategy allows us to include the randomness of the target label sampling. We choose Jester dataset as it generally takes few queries to attack Jester dataset, thus saving testing time. We choose perturbation budget $\rho_{\max} = 16$ as we observe that the attacks under such budget generally take few queries. We choose I3D model because compared to C3D and SlowFast, the attack success rates against I3D are not always 100%; which is good for measuring the error bars for the attack success rates. In addition, compared to TPN, it generally takes fewer queries to launch the attack against I3D. We observe that the gradient estimated by HEURISTICATTACK [6] becomes zero after a certain number of iterations, in which case, no further queries are performed (and hence resulting in a low success rate).

We report the mean, standard deviation, and standard error in Table 3 and present the error bar plot (with mean and standard error) in Figure 1. GEO-TRAP, compared to other methods, requires statistically fewer number of queries while achieving statistically higher attack success rates than the baseline methods.

## D    Additional Experiments with Different Geometric Transformations

GEO-TRAP can employ different kinds of geometric transformations in the TRANS-WARP function. In addition to the translation-dilation transformation ($D = 3$) employed throughout the main manuscript, we report the performance of GEO-TRAP with two other different geometric transformations, i.e., similarity transformation ($D = 4$) and affine transformation ($D = 6$).

Table 2: Additional analysis of attack performance with different perturbation budgets $\rho_{\max}$

| Budget | Methods | Black-box Video Classifiers | | | | | | | |
|---|---|---|---|---|---|---|---|---|---|
| | | C3D | | SlowFast | | TPN | | I3D | |
| | | ANQ ($\downarrow$) | SR ($\uparrow$) | ANQ ($\downarrow$) | SR ($\uparrow$) | ANQ ($\downarrow$) | SR ($\uparrow$) | ANQ ($\downarrow$) | SR ($\uparrow$) |
| **Attack: Untargeted, Dataset: Jester** | | | | | | | | | |
| $\rho_{\max} = 8$ | MOTION-SAMPLER ATTACK [7] | 7310 | 96.3% | 1926 | 100% | 8056 | 91.3% | 5482 | 98.1% |
| | GEO-TRAP (**Ours**) | **2614** | **100%** | **553** | **100%** | **4518** | **92.4%** | **2312** | **100%** |
| $\rho_{\max} = 10$ | MOTION-SAMPLER ATTACK [7] | 4549 | 99.0% | 1906 | 100% | 6269 | 91.3% | 3029 | 99.4% |
| | GEO-TRAP (**Ours**) | **1602** | **100%** | **521** | **100%** | **3315** | **92.4%** | **1599** | **100%** |
| $\rho_{\max} = 16$ | MOTION-SAMPLER ATTACK [7] | 2201 | 100% | 1421 | 100% | 3786 | 96.3% | 1347 | 100% |
| | GEO-TRAP (**Ours**) | **311** | **100%** | **137** | **100%** | **3147** | **96.3%** | **551** | **100%** |
| **Attack: Untargeted, Dataset: UCF-101** | | | | | | | | | |
| $\rho_{\max} = 8$ | MOTION-SAMPLER ATTACK [7] | 16848 | 78.0% | 5436 | 95.0% | 20687 | 70.0% | 9242 | 92.0% |
| | GEO-TRAP (**Ours**) | **12100** | **84.0%** | **2064** | **98.0%** | **18433** | **74.0%** | **6647** | **97.0%** |
| $\rho_{\max} = 10$ | MOTION-SAMPLER ATTACK [7] | 14336 | 81.6% | 4673 | 97.2% | 20369 | 75.8% | 7400 | 94.4% |
| | GEO-TRAP (**Ours**) | **11490** | **86.2%** | **1547** | **98.8%** | **17716** | **76.1%** | **4887** | **97.4%** |
| $\rho_{\max} = 16$ | MOTION-SAMPLER ATTACK [7] | 11605 | 82.0% | 1944 | 99.% | 18055 | 75.8% | 4437 | 96.0% |
| | GEO-TRAP (**Ours**) | **9006** | **86.2%** | **858** | **99.0%** | **15972** | **76.1%** | **2643** | **98.0%** |
| **Attack: Targeted, Dataset: Jester** | | | | | | | | | |
| $\rho_{\max} = 8$ | MOTION-SAMPLER ATTACK [7] | 42136 | 92.6% | 39833 | 98.1% | 121800 | 52.2% | 48788 | 85.2% |
| | GEO-TRAP (**Ours**) | **9333** | **100%** | **11433** | **98.1%** | **51799** | **88.9%** | **25552** | **96.3%** |
| $\rho_{\max} = 10$ | MOTION-SAMPLER ATTACK [7] | 26704 | 98.2% | 33087 | 100% | 63721 | 80.9% | 39037 | 90.7% |
| | GEO-TRAP (**Ours**) | **6198** | **100%** | **7788** | **100%** | **41294** | **92.6%** | **19542** | **98.2%** |
| $\rho_{\max} = 16$ | MOTION-SAMPLER ATTACK [7] | 8696 | 100% | 18901 | 100% | 40643 | 90.7% | 25308 | 94.4% |
| | GEO-TRAP (**Ours**) | **4219** | **100%** | **3855** | **100%** | **16979** | **96.3%** | **9110** | **100%** |
| **Attack: Targeted, Dataset: UCF-101** | | | | | | | | | |
| $\rho_{\max} = 8$ | MOTION-SAMPLER ATTACK [7] | 136327 | 51.7% | 72807 | 76.7% | 153355 | 35.0% | 107304 | 51.1% |
| | GEO-TRAP (**Ours**) | **90401** | **82.5%** | **27306** | **93.0%** | **150052** | **36.8%** | **91773** | **59.3%** |
| $\rho_{\max} = 10$ | MOTION-SAMPLER ATTACK [7] | 100467 | 71.1% | 57126 | 86.0% | 151409 | 31.6% | 96498 | 59.6% |
| | GEO-TRAP (**Ours**) | **71820** | **85.8%** | **21878** | **95.0%** | **141629** | **40.0%** | **76708** | **74.6%** |
| $\rho_{\max} = 16$ | MOTION-SAMPLER ATTACK [7] | 69344 | 79.6% | 37759 | 92.8% | 143504 | 45.0% | 70707 | 75.0% |
| | GEO-TRAP (**Ours**) | **35641** | **98.0%** | **18177** | **95.0%** | **132065** | **45.5%** | **44400** | **86.0%** |

Recall that untargeted attack performance of GEO-TRAP using these three geometric transformations on Jester dataset is reported in the main manuscript (Figure 4). In this section, we present the a more comprehensive set of results on both targeted and untargeted attacks, for both Jester and UCF-101 datasets in Table 4. We observe that the transformation with fewer degrees of freedom, i.e., translation-dilation transformation tends to requires fewer queries while having the same or higher attack success rates on Jester Dataset; this trend is consistent no matter which attack goal is used. On UCF-101 dataset, the transformations with fewer degrees of freedom, i.e., translation-dilation transformation and similarity transformation, require fewer queries while having the same or higher attack success rates compared to the affine transformation.

Table 3: Statistical results with respect to the random seed after running attacks multiple times (*Attack*: Targeted, *victim classifier*: I3D, *Dataset*: Jester, *perturbation budget*: $\rho_{\max} = 16$)

| | Methods | | | | | |
|---|---|---|---|---|---|---|
| | HEURISTIC | | MOTION SAMPLER | | GEO-TRAP | |
| | ANQ ($\downarrow$) | SR ($\uparrow$) | ANQ ($\downarrow$) | SR ($\uparrow$) | ANQ ($\downarrow$) | SR ($\uparrow$) |
| Run 1 | 31088 | 77.9% | 25308 | 94.4% | 9110 | 100% |
| Run 2 | 38388 | 76.0% | 20290 | 96.3% | 10110 | 100% |
| Run 3 | 42098 | 74.1% | 23356 | 94.4% | 5758 | 100% |
| Run 4 | 42022 | 74.0% | 24464 | 96.3% | 7799 | 100% |
| Run 5 | 27431 | 81.5% | 25312 | 94.4% | 11782 | 100% |
| Mean | 36205 | 76.7% | 23746 | 95.2% | 8912 | 100% |
| Standard Deviation | 6643 | 3.1% | 2092 | 1.0% | 2286 | 0% |
| Standard Error | 2971 | 1.4% | 936 | 0.5% | 1022 | 0% |

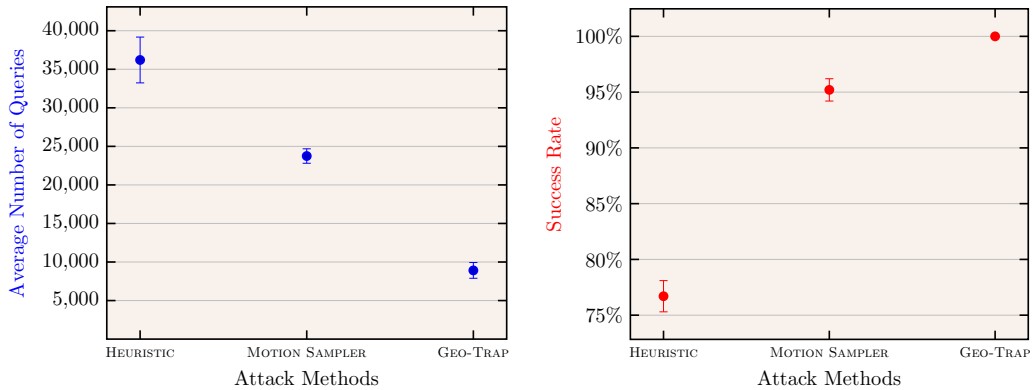

Figure 1: Error bar plot to compare the performance (success rate and average number of queries) of different attack methods. We observe that our method outperforms the baseline methods in a statistically significant way. Detailed numbers are presented in Table 3.

Table 4: Additional analysis of attack performance of GEO-TRAP with different geometric transformations $\mathcal{M}_\phi$

| Geometric Transformations, $\mathcal{M}_\phi$ | Black-box Video Classifiers | | | | | | | |
|---|---|---|---|---|---|---|---|---|
| | C3D | | SlowFast | | TPN | | I3D | |
| | ANQ ($\downarrow$) | SR ($\uparrow$) | ANQ ($\downarrow$) | SR ($\uparrow$) | ANQ ($\downarrow$) | SR ($\uparrow$) | ANQ ($\downarrow$) | SR ($\uparrow$) |
| **Attack: Untargeted, Dataset: Jester** | | | | | | | | |
| Translation | 3340 | 100% | 1316 | 100% | 5305 | 92.4% | 3943 | 100% |
| Dilation | **1407** | **100%** | **325** | **100%** | 3574 | 92.4% | **1239** | **100%** |
| Translation Dilation | 1602 | 100% | 521 | 100% | **3315** | **92.4%** | 1599 | 100% |
| Similarity | 1621 | 100% | 532 | 100% | 3746 | 92.4% | 1629 | 100% |
| Affine | 2716 | 100% | 1057 | 100% | 4579 | 91.6% | 2679 | 100% |
| **Attack: Targeted, Dataset: Jester** | | | | | | | | |
| Translation | 12560 | 100% | 18337 | 100% | 56073 | 83.0% | 46683 | 90.7% |
| Dilation | 6887 | 100% | 8134 | 98.1% | **36898** | **92.6%** | **14019** | **98.2%** |
| Translation Dilation | **6198** | **100%** | **7788** | **100%** | 41294 | 92.6% | 19542 | 98.2% |
| Similarity | 6431 | 100% | 7939 | 100% | 42594 | 90.7% | 19369 | 98.2% |
| Affine | 10326 | 100% | 15360 | 100% | 55276 | 90.7% | 32006 | 94.4% |
| **Attack: Untargeted, Dataset: UCF-101** | | | | | | | | |
| Translation | 13145 | 86.2% | 3959 | 98.0% | 18551 | 3220% | 9078 | 94.0% |
| Dilation | **9991** | **87.6%** | 1510 | **98.9%** | **16847** | **76.7%** | **3755** | **97.4%** |
| Translation Dilation | 11490 | 86.2% | 1547 | 98.9% | 17716 | 76.1% | 4887 | 97.4% |
| Similarity | 10624 | 85.8% | **1489** | 98.6% | 17492 | 76.7% | 5694 | 95.0% |
| Affine | 12792 | 84.8% | 3088 | 98.0% | 17773 | 75.0% | 8291 | 94.0% |

# E    Additional Experiments on GEO-TRAP with Different Loss Functions

In this section, we further validate that, compared to our three baseline methods (i.e., MULTI-NOISE ATTACK [8], ONE-NOISE ATTACK, MOTION-SAMPLER ATTACK [7]), the gradients searched with GEO-TRAP are better. This is demonstrated by the fact that GEO-TRAP's gradients generally have larger cosine similarity with the ground truth gradients. This trend is loss function agnostic, with both untargeted and targeted attacks, as shown in Figure 2. We consider four attack loss functions, three untargeted attack loss functions and one targeted attack loss function, described below.

We start with explaining the flicker loss used for untargeted attack and the cross-entropy loss used for targeted attack in the main paper. Flicker loss is defined with the probability scores of the top-2 labels returned by $f_\theta(x)$ following [9]. In particular, if the attack is not successful, the most likely label predicted by $f_\theta(x)$ will be the true label $y$. We denote the probability score associated with this label as $p_y(x)$. Similarly, we denote the *second* most likely label predicted by $f_\theta(x)$ as $y'$ and its corresponding probability score as $p_{y'}(x)$. The loss function is defined to encourage $p_{y'}(x)$ increasing and $p_y(x)$ decreasing until $p_{y'}(x) > p_y(x)$ and $y'$ becomes the predicted top-1 label. This loss function can be mathematically denoted as follows.

$$\mathcal{L}_{\text{flicker}}(\boldsymbol{x}, y) = \left[ \min\left( \frac{1}{m}\mathcal{K}(\boldsymbol{x}, y)^2, \mathcal{K}(\boldsymbol{x}, y) \right) \right]_+ \text{with}, \mathcal{K}(\boldsymbol{x}, y) = p_y(\boldsymbol{x}) - p_{y'}(\boldsymbol{x}) + m \quad (1)$$

Here, $[a]_+ = \max(0, a)$ and $m > 0$ is the desired margin of the original class probability below the adversarial class probability. We refer readers to [9] for more detailed explanation of (1).

For the targeted attack, the cross-entropy loss is defined as follows.

$$\mathcal{L}(\boldsymbol{x}, y_\top) = -\log\big(p_{y_\top}(\boldsymbol{x})\big) \tag{2}$$

where $p_{y_\top}(\boldsymbol{x})$ is the probability score of the target label returned by $\boldsymbol{f_\theta}(\boldsymbol{x})$.

In addition to the above loss functions, we consider two other untargeted loss functions for gradient analysis of attacks methods. The first one is the untargeted attack loss function defined in [7] based on CW2 loss [10] as shown in the following.

$$\mathcal{L}_{\mathrm{cw}}(\boldsymbol{x}, y) = \big[p_y(\boldsymbol{x}) - p_{y'}(\boldsymbol{x})\big]_+ \tag{3}$$

where, $p_y(\boldsymbol{x})$ is the largest probability score, which should be associated with the true label $y$, and $p_{y'}(\boldsymbol{x})$ is the second largest probability score, which is associated with the second most confident label $y'$. The second loss is a cross-entropy loss where a lower $p_y(\boldsymbol{x})$ is encouraged, as shown in the following.

$$\mathcal{L}_{\mathrm{ce}}(\boldsymbol{x}, y) = -\log\big(1 - p_y(\boldsymbol{x})\big) \tag{4}$$

We calculate the average cosine similarity (over 1000 randomly chosen samples) between the ground truth gradients and the estimated gradients for GEO-TRAP and the three baselines. As shown in Figure 2, for all the five different loss functions considered and on both Jester (see Figure 2(a)) and UCF-101 (see Figure 2(b)) dataset, the gradients searched by GEO-TRAP have better quality consistently. This explains why GEO-TRAP requires less number of queries while achieving the same or higher attack success rates.

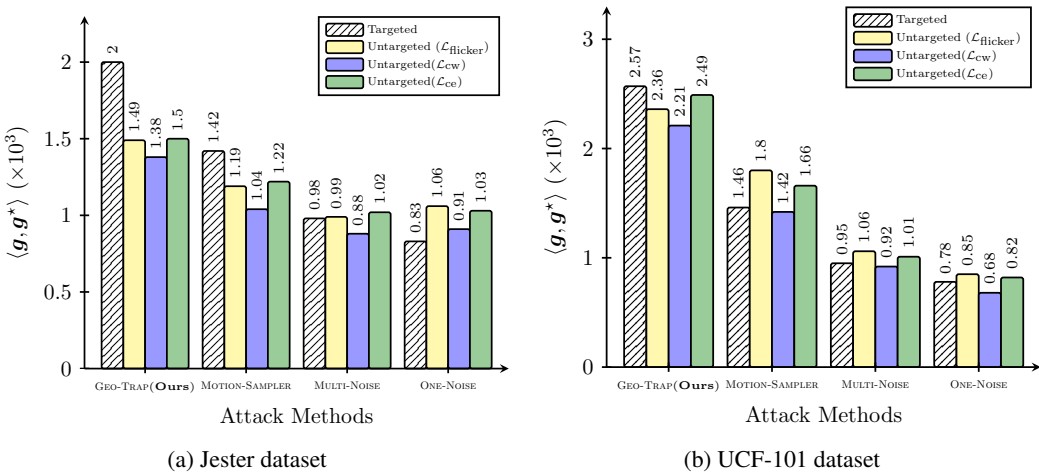

(a) Jester dataset          (b) UCF-101 dataset

Figure 2: Evaluation of gradient estimation quality by calculating the cosine similarity between the ground truth gradient $\boldsymbol{g}^\star$ and the estimated gradient $\boldsymbol{g}$ calculated by different attack methods.

# F   Additional Examples of Adversarial Videos

In this section, we provide additional adversarial examples on both Jester and UCF-101 datasets as shown in Figure 3. We observe that the generated adversarial frames have little difference from the clean ones but can lead to a failed classification.

In addition, we calculate PSNR to measure the perception of perturbations. We measure the minimum PSNR among all frames as it represents the worst-case scenario of maximum degradation for the video. For this, we generate the adversarial examples for untargeted attack against the C3D model on the Jester dataset. The average minimum (across all videos) PSNR of resultant adversarial videos for GEO-TRAP is 28.30 dB; for MOTION-SAMPLER ATTACK [7] is 28.60 dB, and for HEURISTICATTACK [6] is 22.06 dB. We observe that GEO-TRAP, as well as MotionSampler, has less video quality degradation compared to HeuristicAttack.

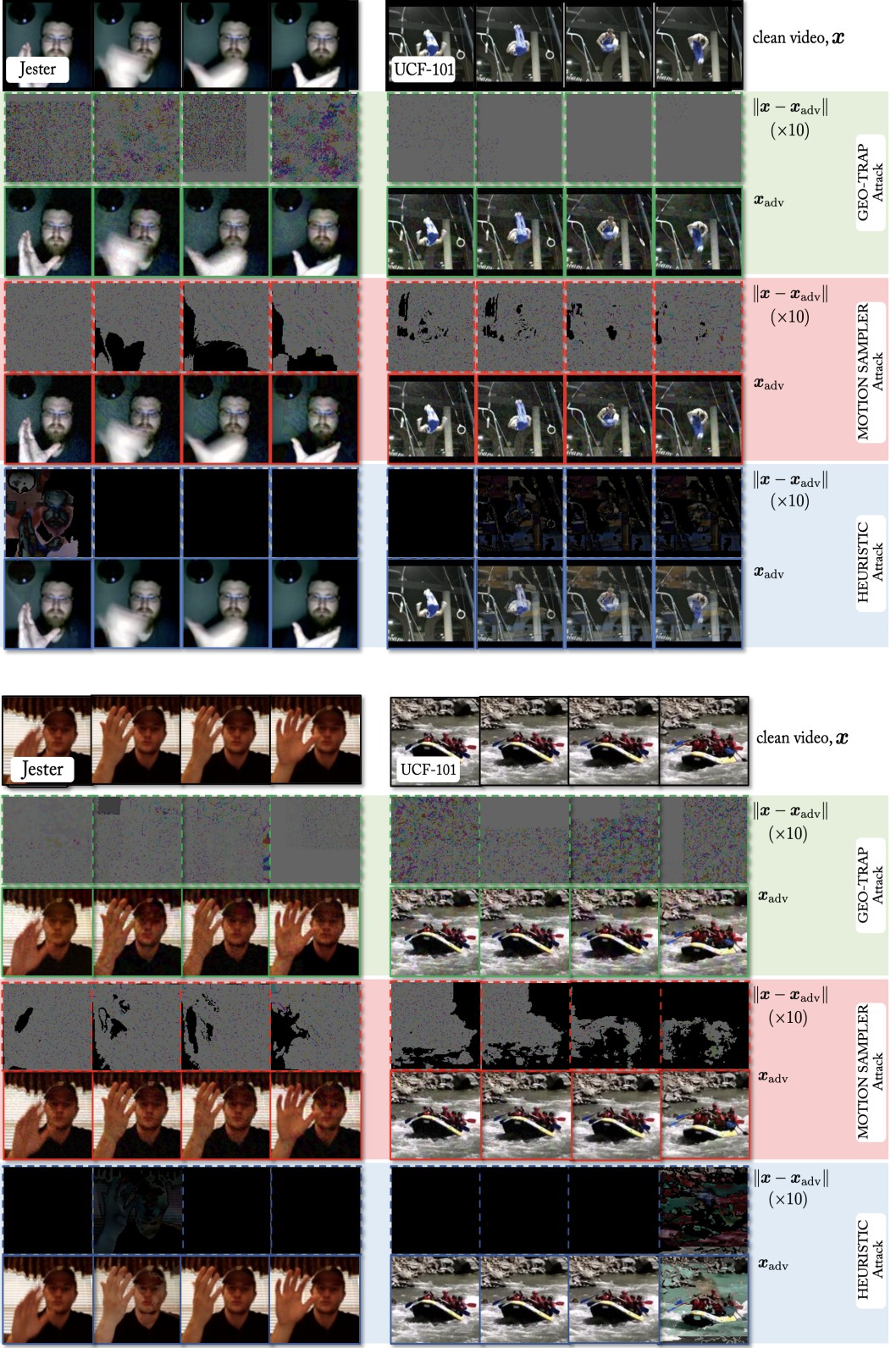

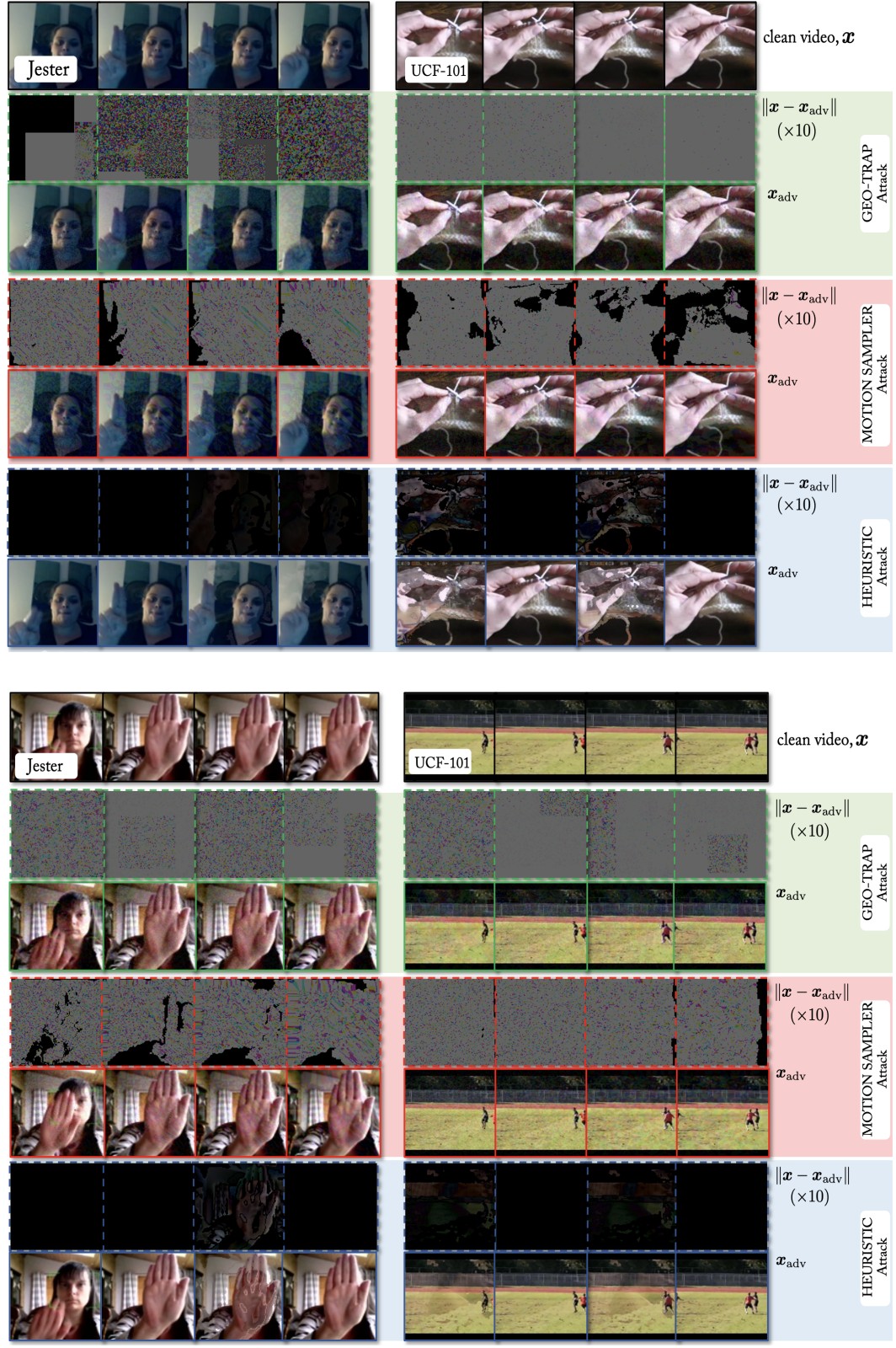

Figure 3: The visualization of the perturbation (×10) and adversarial frames of our methods and the two baseline methods on Jester (left column) and UCF-101 datasets (right column).