# OpenReview forum: "Adversarial Attacks on Black Box Video Classifiers: Leveraging the Power of Geometric Transformations"
_NeurIPS.cc/2021/Conference — NeurIPS 2021 Poster_

### Official Review · Reviewer_zixR · 2021-07-12

**Rating:** 7
**Confidence:** 4

**Summary:**

The paper proposes a query-efficient score-based black-box adversarial attack on a video classification task. To address the problem of the high dimensionality, the paper introduces a simple method to reduce the dimension of the search space by applying random geometric transformations to a single random noise of size $H \times W \times C$ to generate a search direction candidate of size $T \times H \times W \times C$ . The paper empirically shows that the proposed method achieves a higher attack success rate with fewer queries than the baselines on various types of models and datasets.

**Limitations And Societal Impact:**

### Suggestion

- Ablation study: Similar to the experiment in Section 5.2,  I suggest the authors conduct an ablation study that compares the performance of the proposed transformation (translation + dilation) with translation only or dilation only. The authors need to clarify why translation and dilation should be combined.

**Main Review:**

### Strong Points

- Clarity: The paper is well-written and easy to follow. The paper clearly states the threat model and attack settings and specifies the hyperparameters.
- Simplicity: The paper proposes a simple method to generate the effective temporal structure of the search direction with geometric transformations that requires no additional parameters or learning process and can be easily implemented using only matrix multiplications.
- Empirical analysis: The paper provides a thorough analysis of the gradient estimation quality of the proposed method (Figure 2 in Section 4) to validate its effectiveness.
- Experiment: The paper conducts extensive experiments varying datasets, models, and perturbation budgets in both targeted and untargeted settings and shows that the proposed method outperforms the current state-of-the-art methods.

### Weak Points

- Novelty: The proposed algorithm is a simple modification of [1], replacing the motion prior by random geometric transformations. Although this change can significantly improve the attack performance, the novelty of the proposed method is incremental.
- How it works: There is no clear theoretical or empirical explanation of how the proposed method leads to better-estimated gradients.
- Baselines: It seems that the results of [1] are not reproduced properly. Based on the original paper, it can achieve a 98.02% attack success rate with a 4,748 average number of queries against I3D on the UCF-101 dataset, which is different from Table 2 in your paper. Note that this result is comparable to the performance of the proposed algorithm.

### Final thoughts

- The paper proposes a simple method to effectively finds search directions for estimating NES gradients. However, there are critical points on the novelty and reproducing the baseline. Therefore, I marginally accept this paper.

### Reference
[1] Hu Zhang, Linchao Zhu, Yi Zhu, and Yi Yang. Motion-excited sampler: Video adversarial attack with sparked prior. In ECCV 2020

### Update

- Thank you for the clarification. Most of my concerns has been addressed and I raise the score. Please include the results of ablation study (and its empirical analysis) that the performance gain comes mainly from the dilation operation to make the paper stronger.

**Time Spent Reviewing:**

3

---

> ### Author Response · Authors · 2021-08-10
> **Response to Comments by Reviewer zixR**
>
> 1. **Novelty.** As stated previously (Reviewer VM8, Point 3), our main contribution lies in the fact that without pre-computations of strong priors like choosing important frames as in [19] or employing optical flow of the video to add perturbations as in [21] (i.e. reviewer’s [1]), a surprisingly effective strategy of employing well known Geometric Transformations can easily fool diverse video classifiers with few queries on two large as well different kind of video datasets (e.g. on average over 45 % fewer queries than [21]). The intuitive idea (Reviewer VM8, Point 2) of employing such transformations is based on the observation of  object motion being typically governed by such transformations.
> Finally, note that we do not simply replace motion prior by random geometric transformations. Rather, we present a novel strategy to parameterize the important temporal dimension of videos (which video classifiers use to infer their decisions [A]) using geometric transformations, presenting a non-trivial but simple strategy for searching for better gradients. We will make this more clear in the paper.
> &nbsp;
> [A] What Makes a Video a Video: Analyzing Temporal Information in Video Understanding Models and Datasets, CVPR 2018
> ---
> 2. **How it works.** Our hypothesis behind GEO-TRAP leading to better gradients is based on the observation that video classifiers use temporal information for their decisions and object motion in videos is governed by geometrical transformations. However, there are complexities in being able to accurately determine what geometrical perturbation would cause one motion trajectory to be mapped to another.  Such investigation is beyond the scope of this work. We provide a thorough empirical analysis of the gradient estimation quality of the proposed method (Figure 2 in Section 4) to validate the proposed method’s effectiveness. We would explore theoretical analysis in our future works.
> ---
> 3. **Baselines.** Thank you for the question. Kindly note that the original results reported in [19] and [21] do not use the same pre-trained video classifiers as in our work. For example, the recognition accuracy of the I3D model (from MMAction2 library [B]) used in our experiments is 71.7% and that used in [19] is 93.55%. Hence, for a fair comparison, we use the codes released by the authors of [19] and [21] and re-do all the experiments with the same classifiers available to us. We will make this clear in the paper.
> &nbsp;
> [B] OpenMMLab's Next Generation Video Understanding Toolbox and Benchmark, https://github.com/open-mmlab/mmaction2.
> ---
> 4. **Ablation study.** We conduct this ablation study for both targeted attack and untargeted attack on the Jester Dataset for C3D model. Specifically,
> &nbsp;&nbsp;&nbsp;&nbsp;&nbsp;&nbsp; (i) For the untargeted attack, GEO-TRAP with Translation requires 3340 Average Number of Queries (ANQ); GEO-TRAP with Dilation requires 1407 ANQ; GEO-TRAP with both Translation and Dilation requires 1602 ANQ.
> &nbsp;&nbsp;&nbsp;&nbsp;&nbsp;&nbsp; (ii) For targeted attack, GEO-TRAP with Translation requires 12560 Average Number of Queries (ANQ); GEO-TRAP with Dilation requires 6887 ANQ; and GEO-TRAP with both Translation and Dilation requires 6198 ANQ.
> &nbsp;
> In short, we observe that Dilation transformation itself is enough for the untargeted adversarial attack, but the combination of transformation and dilation gives best results when the attack goal is targeted attack. We hypothesize that this behavior could be related to attributes of the Jester Dataset. We will add the ablation study in the paper.

---

> ### Author Response · Authors · 2021-08-25
> **Post rebuttal comment**
>
> Hello Reviewer zixR,
>
> Thank you for updating the score. We will include all the results and discussions from our responses in the paper.
>
> Best wishes,
> Authors

---

### Official Review · Reviewer_VM8Y · 2021-07-15

**Rating:** 6
**Confidence:** 4

**Summary:**

This paper studies adversarial attacks against black-box video classification models. By adopting geometric transformation, the proposed GEO-TRAP reduced the search space for attacking the models. The attack process against black-box models is similar as the extant methods, by estimating the gradient of the loss function. By conducting experiments, the proposed method could use fewer queries to achieve better attack performance.

**Limitations And Societal Impact:**

The paper could consider how to generate such adversarial attacks in real-world, which might be utilized to improve the current classification systems. As adversarial learning with generated adversarial examples might improve the robustness against some (may not exist) adversarial examples, but the accuracy of the original system would be reduced slightly.

**Main Review:**

The paper proposed the GEO-TRAP method to attack black-box video classification models, but the idea seems trivial and it combines many methods from extant attack methods. The experimental results also need some further explanation.
1)	The proposed method adds perturbation to each frame. How to guarantee the continuity of the generated perturbation in different frames?
2)	Please explain the intuitive idea of adopting geometric transformations. How to generate such transformation and evaluate the effectiveness of the transformation by theoretical analysis;
3)	Estimating the gradient for attacking black-box models is commonly adopted in many extant methods. The novelty / main contribution is limited to the dimensionality reduction by randomly sampled geometric transformation parameter tensor (as Section 3.1 stated).
4)	The paper does not state clearly whether the black-box model would output the confidence of the classification result (Section 3). If the model only outputs the classification result (without any confidence value), how to define the loss function? The model settings should be introduced in detail since there are still many different settings of black-box models;
5)	In the experiment, the paper should also compare the added perturbation to the video, not only the success rate and the average number of queries;
6)	The number of maximum queries of the method is set as 60,000 for untargeted attack and 200,0000 for targeted attack. How about the other algorithms? As shown in Table 2 and 3, the success rate of other algorithms is not high; would increasing the maximum number of queries achieves higher success rate?
7)	From the supplementary materials, the classification accuracy of the four black-box models on UCF-101 is lower than that on Jester. The attack success rate on UCF-101 is much higher that that on Jester. The high success rate may lie in low classification accuracy of the original models;
8)	The proposed method adds perturbations to each frame. But such attack method cannot be adopted in real scenario. In many safety-critical systems such as the mentioned perceptual modules in autonomous vehicles, it is difficult to add perturbations on the video.


**Time Spent Reviewing:**

12

---

> ### Author Response · Authors · 2021-08-10
> **Response to Comments by Reviewer VM8Y**
>
> 1. **Continuity of perturbations.** We assume the reviewer suggests temporal continuity of perturbations in different frames. One of our baselines [21] tries to maintain this continuity by using pre-computed optical flow to generate perturbations. However, our experiments (main: Figure 2(b), supp-mat: Table 2,3, Figure 1) demonstrate that such a method to maintain continuity of perturbations in their attack is too restrictive in terms of query efficiency. Since we start with a random tensor $r_{\text{frame}}$ and perform transformation for each frame independently, our method does not result in perturbations that are continuous across frames. However, they outperform other attacks on diverse video classifiers demonstrating such continuity is not essential to get better performance. Thus, we believe that it is acceptable; however, we will examine how to make our query effectual perturbations temporally continuous and the benefits therein in our future work.
> ---
> 2.
> * **Intuitive idea.** This idea behind our choice of Geometric Transformations is that they provide a computationally cheap strategy of transforming a single randomly sampled noise tensor to other candidate directions to search for strong gradients with the following two advantages:
> (i) object motion is typically governed by Geometric transformations _i.e._, movement in the $x$, $y$, and $z$ directions, as well as rotation, and
> (ii) it introduces very few learning/optimizing parameters in generating the attack.
> &nbsp;
> Thus, one can expect these types of transformations to occur in the videos, and this key observation forms the basis for our approach. However, as stated in the paper (L214), there could exist other, potentially better but more complex ways to parameterize the temporal progression of the video perturbation, which is regarded as future work.
> * **How to generate such transformation.** Every geometric transformation is characterized by parameters ($\phi_{11}, \phi_{12}, \cdots$) as shown in Equation (8). The attacker chooses one geometric transformation prior to the attack and randomly samples its parameters at every iteration. E.g. If the attacker chooses Translation-Dilation, then 3 parameters ($\phi_{11}, \phi_{13}, \phi_{23}$) have to be sampled at every iteration. This is explained in Section 3.2 and will be made clearer in the paper.
> * **Effectiveness of transformation.** In Section 4, we have provided an empirical analysis of the effectiveness of our attack method. Developing a theoretical underpinning for why these transformations lead to successful attacks is interesting, but also a hard problem in itself because of the following reason. Although geometric transformations have been analytically understood, perturbations, on the other hand, are data-driven results obtained from computing gradients of video classifiers; thus, these are inherently, significantly much more difficult to theoretically comprehend. It is not the focus of this work and we will consider it in the future.
> ---
> 3. **Novelty.** Our method (as well as our baselines [19, 21]) focuses on reducing the search space of the gradient estimation, and not on the zero-order gradient estimation algorithm.  [19] reduces this by computing a prior defined by keyframes and salient regions that contribute most to the video’s classification decision. Similarly, [21] computes a prior defined by the optical flow of the video to localize regions in the video to add perturbations. However, these methods compute such priors before the attack and hence add a computational burden. GEO-TRAP shows a simple but surprisingly effective strategy of employing well-known Geometric Transformations that can easily fool diverse video classifiers with few queries on two large as well as different kinds of video datasets. We will make this more clear in the paper.
> ---
> 4. **Regarding loss functions.** We follow exact settings of previous video classification attack works [18, 20, 21] that assume the confidence scores are available. The loss function definitions adopted in our method have been described in Section 5 of the supplementary material. We will add an explicit statement pointing to this in the main paper.
> ---
> 5. **Comparison of added perturbations.** Thank you for this suggestion. We followed the exact protocol of previous works [18-21] where the same budget defined by $\Vert\cdot\Vert_p$ norm is applied across all experiments. We have provided a qualitative evaluation in Figure 3 of the main paper and Figure 3 of supplementary material. However, we will update all our quantitative results with PSNR values for all compared attack strategies in the paper. An example is described previously (Reviewer oH6x, Point 1) is the following:  The average (across all videos) minimum PSNR (higher value indicates better quality) of resultant adversarial videos for GEO-TRAP (our method) is 28.30 dB; for MotionSampler [21] is 28.60 dB, and for HeuristicAttack [19] is 22.06 dB demonstrating our attack doesn’t lead to impactful video quality degradation but maintains high attack performance as evidenced in the main paper (Table 2 and 3).
> ---
> 6. **Regarding budget of queries.** This maximum budget is the same budget of queries applied for all the baseline methods as well as GEO-TRAP. We observe that the gradient estimated by [19] could become zero after certain iterations, in which case, no further queries are performed. That is why [19] cannot achieve higher attack success rates by increasing the number of queries. We will add this explanation to the paper.
> ---
> 7. **Relation between success rate and classification accuracy.** We agree with the reviewer here. However, it is important to note that the attack strategy of GEO-TRAP and all the baseline methods are always analyzed on the same video classifier models. Therefore, the experimental results still hold and claims remain validated.
> ---
> 8. **For attacks in real scenarios.** As suggested by [A], digital attacks like our attack method can be applied in a real scenario, for example, the adversary can be a man-in-the-middle that can intercept and add perturbations to streaming video [B], or it could be a previously installed malware that is able to add perturbation prior to classification [C].
> &nbsp;
> [A] Li, S., Neupane, A., Paul, S., Song, C., Krishnamurthy, S. V., Chowdhury, A. K. R., & Swami, A.  Adversarial perturbations against real-time video classification systems. The Network and Distributed System Security Symposium (NDSS) 2018.
> [B] Kaspersky Lab. Defcon,2014. Man-in-the-middle attack on video surveillance systems. https://securelist.com/does-cctv-put-the-public-at-risk-of-cyberattack/ 70008/. [Online; accessed 30-April-2018].
> [C] ZD Net. ZD Net, 2016. Surveillance cameras sold on Amazon infected with malware. https://www.zdnet.com/article/ amazon-surveillance-cameras-infected-with-malware/. [Online; accessed 30-April-2018].

---

### Official Review · Reviewer_AtrB · 2021-07-17

**Rating:** 7
**Confidence:** 4

**Summary:**

This paper proposes a new method for constructing black-box adversarial attacks against video classifiers. Such attacks are usually hard to mount (moreso than attacks against image classifiers) since the dimension of the input is very large (T x C x H x W) instead of just (C x H x W), which increases the complexity of the corresponding gradient estimation problem. Previous work considered reducing this query complexity with various heuristics (e.g., sparsity/only attacking a few frames, or leveraging optical flow)---here, the authors propose an alternative prior. Namely, the proposed method learns a single C x H x W signal, as well as a parameterized (typically affine) transform for each frame (so another T x 6 parameters, for the affine case). This greatly reduces query complexity and is shown to improve upon previous more heuristic priors in practice.

**Limitations And Societal Impact:**

The authors include a Broader Impact section discussing societal impact. As for limitations, as I mentioned above a discussion of the exact scope of the method (in terms of threat models) would strengthen the paper.

**Main Review:**

The work was clearly written and I was able to understand the method rather easily given the text, which is not always the case with black-box attack papers. As far as I am aware, the idea of "splitting" the perturbation signal into a single frame and a temporal transformation is a new and creative one, and the results are enough to convince me that the method achieves a nontrivial improvement over prior work. My main comments are below:

- I thought Figure 1 actually detracted from the readability of the work, particularly because it is called upon before the actual method is introduced in text; I spent a while trying to decipher the Figure when in fact the plain-text description of the algorithm is not so long and is (in my opinion) much easier to follow.

- The paper hypothesizes that the key to GEO-TRAP's improvement is that it leverages the temporal continuity of videos. I think this claim could be strengthen via a quantitative experiment that compares some notion of the "smoothness" or continuity of a video with the query complexity required by each method to attack it. E.g., under the stated hypothesis, we would expect SINGLE-NOISE to do best on the smoothest videos (since it has fewer parameters), MULTI-NOISE to perform best on the least-continuous videos, and GEO-TRAP to perform best on the remaining/middling videos. It would be interesting to understand where the performances intersect and when it is best to use each method.

- In Table 2, the paper should also present numbers of the form "# of queries required to reach X% success rate" for some value of X, or better yet, a graph that puts the query budget on the x axis and the success rate on the y axis. Without this graph it is hard to compare GEO-TRAP with other methods that have smaller query budgets but lower success rates.

- The proposed method seems somewhat specific to L-infinity attacks, since you don't have to care about the interaction between different frames in terms of the perturbation budget. I don't think this is a fatal flaw of the paper, but it would be nice if the authors discussed how the method relies (or does not rely, if the above intuition is mistaken) on the considered threat model).

- Very minor: in table 1, should "perturbation model" be "substitute model"?

**Time Spent Reviewing:**

3

---

> ### Author Response · Authors · 2021-08-10
> **Response to Comments by Reviewer AtrB**
>
> 1. **Position of Figure 1.** Thank you for this suggestion. We will move the figure to a later position in the paper.
> ---
> 2. **Comparison of continuity of a video with the query complexity of attacks.** Thank you for this suggestion. To measure the temporal continuity of videos, we use the $\Vert\cdot\Vert_2$ norm of the frame difference and find the following:
> &nbsp;&nbsp;&nbsp;&nbsp;&nbsp;&nbsp; i) For videos with either low or high continuity in the Jester Dataset, MultiNoise attack always outperforms the OneNoise attack. In particular, the Average Number of Queries (ANQ) for MultiNoise is 5629; ANQ for OneNoise attack is 12477, and ANQ for GEO-TRAP (ours) is 1602.
> &nbsp;&nbsp;&nbsp;&nbsp;&nbsp;&nbsp; ii) For videos with either low or high continuity in the UCF Dataset, OneNoise attack always outperforms the MultiNoise attack. In particular, the Average Number of Queries (ANQ) for MultiNoise is 14148; ANQ for OneNoise attack is 11853, and ANQ for GEO-TRAP is 11490.
> &nbsp;
>  However, we notice that our method GEO-TRAP outperforms both methods in both datasets, demonstrating it can align itself to the temporal progression of the video and as well perform with fewer queries. As pointed out by the Reviewer, we conjecture that this is because of the motion information that plays an important role in the video classification, also noted in [A, B]. Specifically, the videos in the UCF dataset can be distinguished by only the appearance information. However, in the Jester dataset, the appearance of different videos is similar (e.g. different hand gestures videos). Therefore, more motion information needs to be perturbed to fool the classifiers trained on the Jester dataset, and thus MultiNoise attack outperforms the OneNoise attack. Similarly, not much motion information is needed to be perturbed to fool the classifiers trained on the UCF dataset, and hence, OneNoise attack with a smaller search space is enough and outperforms the MultiNoise attack.  We will add this explanation to the paper.
> &nbsp;
> [A] SlowFast Networks for Video Recognition, ICCV 2019.
> [B] ​​What Makes a Video a Video: Analyzing Temporal Information in Video Understanding Models and Datasets, CVPR 2018.
> ---
> 3. **Graph from Table 2.** Thank you for the suggestion. We will convert our results into graphs for better understandability in the paper.
> ---
> 4. **Specificity of GEO-TRAP.** Thank you for the suggestion. We follow previous works [18-21] and use $\Vert\cdot\Vert_\infty$ to do the attack. Our contribution is reducing the search space of the gradient estimation by using the geometric transformation, which is agnostic to the gradient estimation algorithm (which depends on the type of $\Vert\cdot\Vert_p$ attack). Consequently, as stated in [12], the gradient estimation algorithm used in our paper can be applied to other $\Vert\cdot\Vert_p$ norms. Hence, GEO-TRAP can be employed with other $\Vert\cdot\Vert_p$ attack budgets as well.
> ---
> 5. **Regarding Table 1.** The ‘perturbation model’ for [18] acts as a substitute network for the victim classifier in order to perform the attack, and hence acts as a ‘substitute model’ for the attack methods. The ‘perturbation model’ with respect to [20] is a perturbation generation model based on reinforcement learning. To make these concise, we used the term “perturbation model”. We will make this clear in the paper.

---

### Official Review · Reviewer_oH6x · 2021-07-20

**Rating:** 5
**Confidence:** 2

**Summary:**

This paper proposed a block-box adversarial attack method on the video classification task. The proposed method requires fewer queries with high attack success rates compared to SOTA methods, by leveraging Geometric transformations to parameterize and reduce the search space.

**Limitations And Societal Impact:**

see the main review.

**Main Review:**

This paper is well written and organized. However, I have some concerns about the experimental results:

1. from fig.3 the proposed method show perception video quality degradation compare with the clean video. what are the numerical differences (like PSNR) between x and xadv on different attack methods? Will this attack impact the video quality?

2. from Table 2 and Table 3, why does [19] have less ANQ compare with the proposed method at the UCF-101 dataset on C3D and TPN classifier?
It would be better if the authors would provide detailed and insightful explanations.

3. what's the exact time to finish an attack of the proposed method compared with the other SOTA methods instead of the number of query reduction?


**Time Spent Reviewing:**

2 hours

---

> ### Author Response · Authors · 2021-08-10
> **Response to Comments by Reviewer oH6x**
>
> 1. **Response to Point 1.** In [18-21], $\Vert\cdot\Vert_p$ norm is used to measure the perception of the perturbation/video quality degradation. We follow the same spirit in our experiments (L237-240). Nonetheless, we measure the minimum PSNR among all frames as it represents the worst-case scenario of maximum degradation for the video. For this, we generate the adversarial examples for untargeted attack against the C3D model on the Jester dataset. The average minimum (across all videos) PSNR of resultant adversarial videos for GEO-TRAP (our method) is 28.30 dB; for MotionSampler [21] is 28.60 dB, and for HeuristicAttack [19] is 22.06 dB. We observe that GEO-TRAP, as well as MotionSampler, has less video quality degradation compared to HeuristicAttack. As observed from the high PSNR values, it can be concluded that our attack doesn’t lead to impactful video quality degradation, but maintains high attack performance as evidenced in the main paper: Table 2 and 3.
> ---
>
> 2. **Response to Point 2.** Thank you for this insightful question. We observe that the gradient estimated by [19] becomes zero after a certain number of iterations, in which case, no further queries are performed (and hence resulting in a low success rate). We believe this occurs because of the following key property of [19]’s attack strategy. Before the attack optimization, [19] first computes a prior by searching for a subset containing the most important frames of the video (in order to perturb this subset rather than all frames) that contribute most to its classification decision. Further, as TPN uses multi-scaled features extracted by I3D as the backbone (requiring reasonable multi-scale features from the subset) and C3D focuses on mostly on salient motion (requiring reasonable temporal features from the subset), the aforementioned prior computed by [19] results in gradients that show good performance with a fine-grained dataset like Jester (rapid change in frames) but poor performance with a coarse-grained dataset like UCF101 (minimal change in frames).
> ---
>
> 3. **Response to Point 3.** We follow [18-21] and use the number of queries to measure the efficiency of the attack method. The number of queries matters because:
> &nbsp;&nbsp;&nbsp;&nbsp;&nbsp;&nbsp; i) too many similar queries could trigger the alarm and be detected by any defense system.
> &nbsp;&nbsp;&nbsp;&nbsp;&nbsp;&nbsp; ii) most servers have limitations on how many queries a client could make during a certain time period.
> The exact attack time is not a metric of common consideration in previous works [18-21]; to be consistent, we adhere to the number of queries as our metric. Nonetheless, we measure the attack time with the C3D model acting as the victim model on Jester dataset (untargeted attack) with the hardware configuration: GeForce RTX 2080 (RAM size is 256GB) and software configuration: Linux OS, Python-based code, and Pytorch as DL framework. GEO-TRAP takes 33 mins, Heuristic attack [19] takes 76 mins, and MotionSampler [21] attack takes 91 mins (note that this does not include the time taken by [21] to calculate the optical flow of the videos). It clearly demonstrates the advantage of our method over others in terms of the time required to finish the attack.

---

### Author Response · Authors · 2021-08-10
**Official Response**

We are very thankful to the reviewers for their thorough reviews and insightful comments. Some of the clarifications sought will truly improve the quality of our paper. Below are our detailed responses.

---

> ### Author Response · Authors · 2021-08-25
> **Discussion on further concerns**
>
> Hello Reviewers,
>
> Thank you again for your helpful and constructive comments. We would be happy to address any further concerns you have based on our responses. Looking forward to further discussions.
>
>
> Best wishes,
> Authors

---

> ### Author Response · Authors · 2021-09-01
> **Discussion on further concerns**
>
> Dear Reviewer oH6x and Reviewer VM8Y,
>
> Hope this letter finds you well.
>
> Since you haven't responded to our rebuttal yet, we wanted to kindly ask you to take some time to go through the replies that we have carefully prepared. Let us know if you have any concerns or confusion, we are happy to clarify. We hope to take full advantage of this discussion phase to eliminate misunderstandings, and there is not much time left.
>
> Being a reviewer is a privilege that requires a significant investment of time, and we understand this may be in conflict with your other duties. We appreciate every second you put into evaluating our work, and we hope you can take a more active role in this reviewer assignment. Your view is important to us, we want to make sure all your concerns have been adequately addressed. We can not respond if we do not have any feedback, please let us know.
>
> Best Wishes,
>
> Authors

---

> > ### Comment · Reviewer_VM8Y · 2021-09-02
> > **Some concerns about the paper**
> >
> > Thanks for the authors' response and I have read all the response to the reviewers. Considering my review comments, I still have some concerns:
> > 1) Is temporal continuity of perturbations is needed in generating adversarial video? In my view, such temporal continuity can make the video  more natural. Otherwise, generating adversarial video can be regarded as generating multiple images (each frame in the video);
> > 2) The fairness of comparison. [21] maintains the continuity of perturbations,while the proposed method does not. Such comparison seems unfair;
> > 3) The added perturbations of the method seems worse than [21]. It is better to introduce the impact of added perturbations.

---

> > > ### Author Response · Authors · 2021-09-02
> > > **Response to further concerns**
> > >
> > > Thank you for your time to go through our response and raise your concerns. All the comments that you (and the other reviewers)  have provided have significantly improved our paper’s quality. Below are our responses:
> > >
> > > ---
> > > **Response to Point 1.** As the Reviewer points out, the goal of all adversarial attacks is to make the perturbations imperceptible to humans in addition to fooling the target model. Hence, the perturbed pixels need not match the temporal flow of the video, i.e., because the perturbations from the successful attack are well hidden (See the human study presented in the last point). Moreover, temporal continuity of video perturbations is not necessary, as shown in previous works [13-20]. Finally, our experiments (Table 2, 3) show that enforcing temporal continuity (as in [21]) is comparatively inefficient.
> > >
> > > ---
> > > **Response to Point 2.** Maintaining the continuity of perturbations is not a condition to subvert classification decisions and in fact, we believe that the very reason that [21] chooses a continuous perturbation contributes to its lower effectiveness. [21] is considered a baseline as it presents an approach to attack video classifiers by searching for better gradients in order to compute effective perturbations. As we show in our paper, maintaining the continuity of perturbations (in [21]'s case, using optical flow) requires more queries than our approach (Table 2 and 3). Kindly note that even our other baseline [19] doesn’t consider this continuity.
> > >
> > > ---
> > > **Response to Point 3.** We would like to first emphasize that the perturbation added by GEO-TRAP (Ours) is as imperceptible as the MotionSampler [21] attack. We did a simple human evaluation and asked 10 individuals to visualize 10 adversarial video clips generated by our method GEO-TRAP and [21]. The 10 adversarial video clips are the ones that are presented in Figure 3 in the main paper and Figure 3 in the supplementary material. 63% of the cases people think the perturbation generated by GEO-TRAP is more imperceptible. We will include the human study in the paper. Also, note that we outperform [21] as demonstrated in Tables 2 and 3 in terms of attack success rate and the average number of queries. The perceptibility of perturbations is controlled by the budget imposed during the attacks. We align with previous works and use $\Vert\cdot\Vert_\infty$ norm to set this budget. We have also provided PSNR values in our response below.
> > >
> > > ---
> > > Please feel free to further ask questions based on the above points.

---

> > > > ### Comment · Reviewer_VM8Y · 2021-09-02
> > > > **About the continuity of perturbations**
> > > >
> > > > I agree some of your response. But if the (temporal) continuity is not necessary, what is the difference of generating adversarial example to each frame of the video? Second, I agree that considering continuity makes [21] less efficient, since it adds some extra constraint in adding perturbations. If the method of [21] does not consider continuity, would it requires less queries? Such comparison would be fair when you remove the continuity constraint in [21].

---

> > > > > ### Author Response · Authors · 2021-09-02
> > > > > **Response to Continuity of perturbations**
> > > > >
> > > > > Thank you for your quick response and further queries.
> > > > >
> > > > > ---
> > > > > ***But if the (temporal) continuity is not necessary, what is the difference of generating adversarial example to each frame of the video?***.
> > > > >
> > > > > Continuity and correlation are different things. [21] is the only work that uses the optical flow and gets temporal continuity as far as we know. Note that temporal continuity is not the goal of [21], but a resultant attribute of employing optical flow, which we explain in the next point. While the perturbation may not be temporally continuous, those applied on the various frames are correlated, as shown in previous works [13-20] (includes white-box video attacks). It is because correlation across the perturbations is what is important -- not temporal continuity. Independently perturbing the different frames will not lead to the misclassification of the activity as desired. This correlation in GEO-TRAP is achieved by the TRANS-WARP function.
> > > > >
> > > > > ---
> > > > > ***If the method of [21] does not consider continuity, would it requires less queries?***
> > > > >
> > > > > The whole premise of GEO-TRAP, and baselines [19, 21] is how an attack can search for potent gradients by creating a better search space. The very reason that [21] uses optical flow is not to maintain temporal continuity but to reduce this search space. Similarly, [19] creates this search space by computing important frames first. With their respectively defined search space, GEO-TRAP as well as [21, 19] employ zero-order optimization to search for the gradients in order to create perturbations.
> > > > >
> > > > > If no optical flow is used, then [21] degrades to its own baselines, that is, One-Noise attack or Multi-Noise attack (shown in their paper’s Table 3). Further, as shown in Section 4 of our paper, we outperform these [21]’s baselines as well. Hence, as continuity is an attribute of the attack method of [21], the comparison is fair since the attack objective remains the same.
> > > > >
> > > > > ---
> > > > > Please feel free to further raise your questions!

---

> > > > > > ### Comment · Reviewer_VM8Y · 2021-09-03
> > > > > > **Update the score**
> > > > > >
> > > > > > The response seems reasonable but I'm not totally convinced by the comparison (about the settings and fairness). I only raise the score to 6.

---

### Decision · Program_Chairs · 2021-09-27

**Decision:**

Accept (Poster)

**Comment:**

The authors propose a black-box adversarial attack method for video classifiers. The challenge is to reduce the search space (T x C x H x W) for attack efficiency measured in number of queries. The idea is to employ geometrics transformations to parameterize and reduce the search space. The experiments demonstrate significant improvement in higher success rate at fewer queries compared to the baseline methods. I suggest the authors incorporate the suggestions raised by the reviewers in the camera ready version. Please include a) plot of query budget versus success rate in addition to the tables 2 & 3; b) the ablation study for the results on translation only and on dilation only.